# FuseAgent: A VLM-driven Agent for Unified In-the-Wild Image Fusion

## Abstract

Fusing multi-source images captured in the wild is often undermined by unpredictable and coupled degradations, including pixel-level misalignment, adverse weather, and dynamic artifacts. Existing solutions face notable limitations: (1) Task-specific models rely on predefined degradation priors and fail to generalize to the complex, coupled degradations present in real-world scenarios. (2) All-in-one methods, while designed for multi-fusion tasks, frequently overlook the degradation inherent in input images, leading to suboptimal performance. To address these challenges, we introduce FuseAgent, a VLM- powered agent system that autonomously identifies degradations in the input images and dynamically coordinates expert models to execute a tailored fusion strategy. FuseAgent undergoes a two-stage training process: an initial supervised fine-tuning (SFT) establishes basic degradation perception and tool-use skill, followed by Group Relative Policy Optimization for fusion (GRPO-F) augmented with multi-dimensional rewards to further enhance its decision-making and tool proficiency. Experimental results demonstrate the superior performance of FuseAgent in handling complex and coupled degradations in real-world scenes, achieving a **20%** average improvement across all evaluation metrics on challenging in-the-wild benchmarks.

## 1 Introduction

Image fusion is critical for vision-centric applications—including autonomous driving, computational photography, and remote sensing—where multi-source images acquired in real-world scenarios often suffer from complex and intertwined degradations such as pixel misalignment, adverse weather conditions, and dynamic scene artifacts. As shown in Figure 1, existing solutions can be roughly classified into two categories: 1) Task-specific models show proficiency in individual degradation types, such as geometric deviations Huang et al. (2022); Wang et al. (2022) and motion artifacts Cao et al. (2023); Chen et al. (2025), but struggle with the dynamic and coupled degradations in real-world scenarios. 2) All-in-one approaches attempt to consolidate several fusion tasks into a single pipeline, yet frequently neglect the intrinsic unpredictable and mixed degradations of the input images, thereby limiting their practical applicability. To mitigate these challenges, an intuitive solution—shown in Figure 1(a)—is to combine expert models into a fixed pipeline. Yet two fundamental challenges persist: 1) Dynamic degradations, where inputs are affected by complex and coupled distortions that differ across samples, thereby undermining the effectiveness of predefined pipelines. 2) Task adaptability, where identifying the optimal sequence of expert models for each input is still an open challenge.

Recently, the rapid advancement of Large Language Models (LLMs)—which exhibit remarkable capabilities in reasoning, decision-making, and interacting with diverse environments (Patil et al., 2023; Jain, 2022; Yang et al., 2024; Zhao et al., 2025)—has inspired us to rethink image fusion from an agentic perspective. Specifically, an intelligent fusion agent could be *interpreting scene-specific degradations, coordinating expert competencies*, and *assembling optimal fusion workflows*. However, realizing such an agent in practice demands addressing three key challenges: 1) The scarcity of training data, particularly paired clean/degraded images and annotated expert sequences. 2) The difficulty of selecting suitable expert models within an enormous combinatorial space, and 3) The lack of infrastructure designed for training agentic fusion systems.

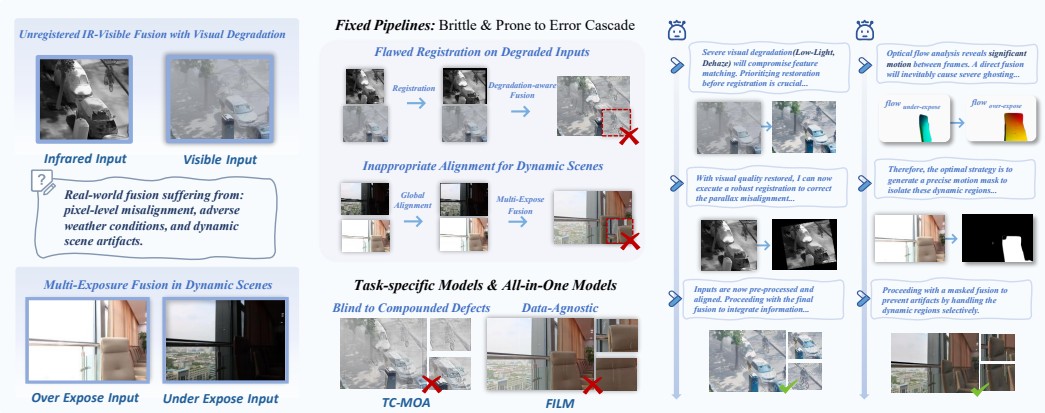

Figure 1: Limitations of task-specific methods, all-in-one methods, and human-defined processing pipelines. 1) Single-task and all-in-one methods fail to effectively handle coupled degradations in real-world scenarios. 2) Static processing pipelines, predefined by humans using multiple expert models, show promise but struggle with dynamic degradations. Moreover, the optimal task sequence remains unpredictable in real-world conditions. In contrast, FuseAgent can dynamically assess scene-specific degradations, coordinate expert models, and plan optimal fusion workflows, yielding superior fusion results.

To translate this agent-driven paradigm into a practical system, as depicted in Figure 1, we establish FuseAgent by employing a Vision-Language Model (VLM) as the agent's core controller, endowing it with degradation perception, decision-making, and tool-use skills. We adopt a two-stage training regime: (1) Supervised Fine-Tuning (SFT) initially teaches the agent foundational skills, including degradation identification and tool invocation, using a curated set of expert trajectories. (2) Group Relative Policy Optimization for Fusion (GRPO-F) is applied to boost system generalization, reduce hallucinations, and improve decision-making in real-world situations. To ensure stability during the GRPO-F process, we introduce multi-dimensional rewards designed for tool-integrated fusion tasks: the Intrinsic Quality Reward (IQR) and the Relational Quality Reward (RQR). The IQR evaluates the perceptual quality of each fused image generated during the rollout, while the RQR quantifies the improvement in "fusion compatibility" between the source images and the fused output. These rewards—inter-image (IQR) and intra-image (RQR)—provide essential, unsupervised guidance for the agent's evolution throughout the process.

Our main contributions are as follows:

- We introduce a novel fusion paradigm, FuseAgent, a VLM-powered agent that autonomously identifies degradations and coordinates specialized expert models to effectively handle complex and coupled degradations in real-world environments.

- We propose a novel two-stage framework combining SFT and GRPO-F, augmented with task-specific rewards—Intrinsic Quality Reward (IQR) and Relational Quality Reward (RQR)—to enable label-free reinforcement learning and improve generalizability in real-world settings.

- Comprehensive experiments demonstrate that FuseAgent excels in both image fusion quality and task-level decision-making, significantly outperforming advanced all-in-one baselines and static, manually designed pipelines.

## 2 RELATED WORK

**Task-specific Image Fusion.** To bridge the gap between idealized fusion assumptions and the complexities of real-world inputs, a significant body of research has focused on task-specific robust fusion. Early efforts concentrated on geometric robustness, leading to registration-free models (Arar et al., 2020; Wang et al., 2022; Xu et al., 2022). While effective at bypassing explicit registration, their narrow specialization renders them ineffective against photometric or dynamic inconsistencies. Another major line of research targets dynamic scenes, proposing specialized deghosting al-

gorithms (Cao et al., 2023; Chen et al., 2025) to mitigate motion artifacts. However, these methods are often predicated on the assumption of well-registered frames with consistent illumination, limiting their applicability in more varied scenarios. More recently, methods have emerged to tackle perceptual quality degradations. Approaches like Text-IF (Yi et al., 2024) and DRMF (Tang et al., 2024) aim to perform fusion while simultaneously addressing issues such as sensor noise, low resolution, or poor illumination. Despite their advances in joint restoration and fusion, they typically presuppose a static, pre-aligned scene. Collectively, while these task-specific approaches demonstrate expertise on isolated problems, they lack a unified framework to address the compounded and heterogeneous degradations endemic to in-the-wild data, as they are incapable of reasoning about the complex interplay of multiple, co-occurring defects.

**All-in-One Image Fusion.** All-in-one fusion paradigm aims to create a single, unified framework for diverse fusion tasks, moving beyond the single-problem focus of specialized models. Early works in this area concentrated on designing unified network architectures, employing powerful backbones like Transformers (Ma et al., 2022) or carefully designed objectives (Xu et al., 2020) to create a shared representation for multiple tasks. More recent approaches have evolved towards greater flexibility. One prominent trajectory leverages generative models to reframe fusion as a conditional synthesis task (Liang et al., 2022). Concurrently, another direction has focused on adaptive mechanisms, such as textual prompts (Zhao et al., 2024; Cao et al., 2025) or modular adapters (Zhu et al., 2024b; Cheng et al., 2025), to guide a core model toward specific fusion objectives. Despite their architectural diversity and task versatility, these all-in-one paradigms share a common limitation: their operational logic remains static. They lack an explicit perception and planning capability to dynamically react to unforeseen, input-level degradations (e.g., misalignment, adverse weather), rendering their fixed frameworks insufficiently robust for the challenges of in-the-wild data.

**VLM-powered Agents.** The limitations of static paradigms in vision tasks have motivated a shift toward dynamic, agent-based systems, a trend fueled by the recent success of Large Language Models (LLMs) in complex reasoning and tool use (Qin et al., 2023; Shen et al., 2024). Foundational works have demonstrated that LLMs and VLMs can be adapted into powerful controllers, capable of orchestrating diverse external tools and foundation models to solve complex user queries (Patil et al., 2023; Wu et al., 2023a). This agent-based paradigm is now being explored in low-level vision, particularly for image restoration. Current approaches in this area can be broadly categorized. One line of research utilizes powerful, off-the-shelf VLMs as zero-shot planners to coordinate restoration tools based on commonsense reasoning (Zhu et al., 2024a; Bai et al., 2025a). Another category focuses on fine-tuning VLMs on domain-specific, often synthetic, data to create expert controllers that can generate explicit, step-by-step execution plans (Chen et al., 2024). While these pioneering efforts validate the potential of visual agents, they have been exclusively focused on single-image restoration. Our work introduces this paradigm to the fundamentally different domain of multi-image fusion. This presents a unique set of challenges, requiring the agent not only to identify degradations but also to reason about complex inter-image relationships, such as geometric alignment, photometric consistency, and complementary information. To address this, FusionAgent is explicitly designed with a training methodology (SFT+GRPO-F) and a relational reward (RQA) tailored for learning these complex relational policies from unlabeled in-the-wild data.

## 3 METHODOLOGY

In this section, we first describe the overall workflow of FuseAgent (Sec. 3.1). Next, we introduce our data generation pipeline, which constructs a large-scale dataset, including expert reasoning traces and sequences for agentic fusion tasks. Finally, we detail the two-stage training framework for FuseAgent, comprising supervised fine-tuning and group relative policy optimization for fusion in an unsupervised manner (Sec. 3.3).

### 3.1 OVERVIEW

FuseAgent is a VLM-powered agent system designed for unified in-the-wild image fusion. As shown in Figure 2, FuseAgent autonomously perceives degradation, decomposes complex fusion tasks into subproblems, and assigns the most suitable expert model to each subtask. The pipeline comprises three principal stages: (1) **Perception and task decomposition**, where the agent first analyzes the input image pair to identify a complex set of coupled degradations, such as misalign-

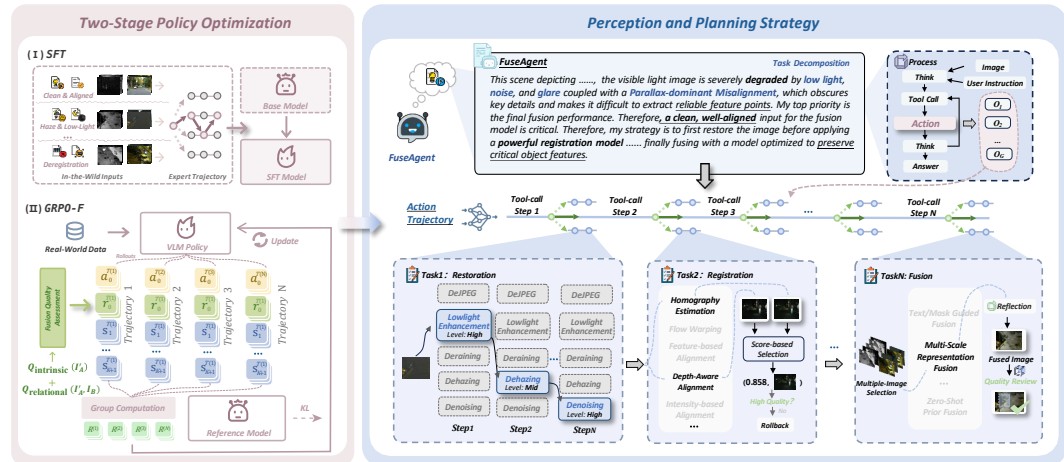

Figure 2: Overview of the two-stage optimization framework for FuseAgent. Initially, FuseAgent undergoes supervised fine-tuning (SFT) on expert-annotated trajectories to establish foundational skills in degradation perception and workflow planning. Following this, the Group Relative Policy Optimization for Fusion (GRPO-F) algorithm is applied to further enhance FuseAgent's decision-making, tool proficiency, and robustness on unlabeled, in-the-wild data.

ment, low light, and dynamic artifacts. (2) **Reasoning and planning**, where the agent formulates multi-step fusion solutions referred to as the "Action Trajectory," based on its expert-level fusion knowledge, determining task sequences and selecting the corresponding expert models. (3) **Tool execution**, where the agent executes the planned task and expert tool sequence to obtain the final fused results. Formally, FuseAgent defines a function as follows:

$$f(I_A, I_B) \rightarrow \mathcal{A} = \{a_1, a_2, \dots, a_n\},$$

where $I_A$ and $I_B$ are the source images, and $\mathcal{A}$ is the planned Action Trajectory. Each action $a_i$ represents a specific tool-call (*e.g.*, Register($I_A$, $I_B$) or Dehaze($I_A$)). The final fused image is obtained by $I_{\text{fused}} = g(I_A, I_B, \mathcal{A})$, where $g(\cdot)$ represents the tool execution environment.

## 3.2 Data Generation Pipeline

**Stage I: Generation of image pairs.** Our data generation process employs a hybrid strategy, combining authentic degraded image pairs from public benchmarks with systematically synthesized data. While existing real-world datasets provide a crucial foundation of authenticity, we observe that they predominantly feature isolated or single-type degradations (e.g., containing motion artifacts or misalignment, but rarely both). To bridge this gap and ensure our agent is trained and evaluated on the compounded challenges truly representative of in-the-wild conditions, we augment this real data by synthesizing complex, multi-defect scenarios. This is accomplished using a modular degradation library capable of introducing and layering a wide range of fusion-specific defects. These include not only adverse weather and sensor noise, but also challenging geometric misalignments, photometric inconsistencies (e.g., exposure variations), and dynamic artifacts. This hybrid approach yields a dataset that captures a comprehensive spectrum of challenges, from simple, single-defect cases to complex, multi-degradation scenarios.

**Stage II: Generation of expert responses (CoT & Actions).** For each degraded image pair, we generate the corresponding expert response, which consists of two parts. (a) **CoT rationales**: To generate detailed reasoning, we leverage a powerful low-level vision VLM. Given a degraded pair, we prompt the model to perform a step-by-step analysis, identifying each degradation, explaining its potential impact on fusion, and verbalizing a high-level strategy. (b) **Optimal action trajectory**: To determine the ground-truth sequence of tool calls, we employ an exhaustive search strategy. We explore all viable permutations of applicable tools and model combinations, scoring each complete trajectory using a set of relational quality metrics to identify the optimal path $\mathcal{A}$. A comprehensive description of this pipeline is provided in Appendix B.

### 3.3 FUSEAGENT FRAMEWORK

The core of FuseAgent is a VLM-based policy network, $\pi_\theta$, which is trained to generate an optimal action trajectory. We cultivate its expert-level capabilities via a two-stage optimization process.

#### 3.3.1 SUPERVISED FINE-TUNING (SFT)

Following recent agent development paradigms (Guo et al., 2025; Chen et al., 2024), we first initialize the policy via Supervised Fine-Tuning (SFT) on our generated dataset. This phase instills foundational skills in degradation identification and tool invocation by teaching the agent to mimic expert trajectories. The SFT objective is to maximize the likelihood of the expert's response $\mathcal{R} = \{\mathcal{C}, \mathcal{A}\}$, optimized via a standard cross-entropy loss:

$$\mathcal{L}_{\text{SFT}}(\theta) = -\sum \log \pi_\theta(\mathcal{R}|I_A, I_B). \tag{1}$$

#### 3.3.2 REINFORCEMENT LEARNING FOR ROBUST POLICY MAKING

Building on the SFT-initialized policy, we introduce a reinforcement learning (RL) stage to enhance the agent's robustness and generalization on unlabeled, in-the-wild data. We employ Group Relative Policy Optimization for Fusion (GRPO-F) (Shao et al., 2024; Jaech et al., 2024), a policy-gradient algorithm well-suited for this task. Refer to Appendix A.1 for further details on the implementation. The key to our approach is a novel, unsupervised reward signal that is action-dependent, distinguishing between intermediate pre-fusion steps and the final fusion action.

For any intermediate pre-fusion action $(a_t, t < n)$, the reward $R_t$ is designed to assess its contribution to improving the conditions for the final fusion. It is a composite signal:

$$R_t = w_i \cdot R_{\text{intrinsic}} + w_r \cdot R_{\text{relational}}.$$

where $w_i$ and $w_r$ are balancing weights.

**Intrinsic quality reward.** This reward, $R_{\text{intrinsic}} \in [0, 1]$, assesses the standalone perceptual quality of an action's output image, $I'_t$. Its primary role is to encourage effective restoration and prevent the introduction of new artifacts. It is formulated as a weighted composite score from a suite of robust, no-reference IQA metrics:

$$R_{\text{intrinsic}}(I'_t) = \sum_{m \in M_{\text{IQA}}} w_m \cdot \hat{\Phi}_m(I'_t). \tag{2}$$

Here, $M_{\text{IQA}}$ is the set of selected IQA metrics, $w_m$ is a predefined weight for each metric $m$ to balance its contribution, and $\hat{\Phi}_m$ is the normalized score produced by that metric.

**Relational quality reward.** The key of our reward design is $R_{\text{relational}}$, a metric that quantifies the improvement in "fusion compatibility" between the source images. It provides a critical guidance signal for all pre-fusion steps. As ground-truth is unavailable, it is defined as a multi-dimensional score evaluating the gain in inter-image consistency. We provide a detailed formulation in Appendix A.2.

$$R_{\text{relational}} = \sum_{d \in D} w_d \cdot \mathcal{C}_d(I'_t, I_t, I_{\text{other}}). \tag{3}$$

where $D$ is the set of predefined compatibility dimensions, $w_d$ is the corresponding weight for each dimension $d$, and $\mathcal{C}_d$ is the compatibility score function. The inputs to this function are the output image $I'_t$, the original image before the action $I_t$, and the other unprocessed image in the pair, $I_{\text{other}}$.

## 4 EXPERIMENTS

### 4.1 EXPERIMENTAL SETUP

**Dataset Setting.** To validate performance under authentic in-the-wild conditions, we curated a comprehensive evaluation benchmark from a variety of public datasets. This benchmark was purposefully constructed to span a wide spectrum of real-world challenges, ranging from clean, well-controlled scenes to complex scenarios featuring multiple, coupled degradations. Our scenarios are

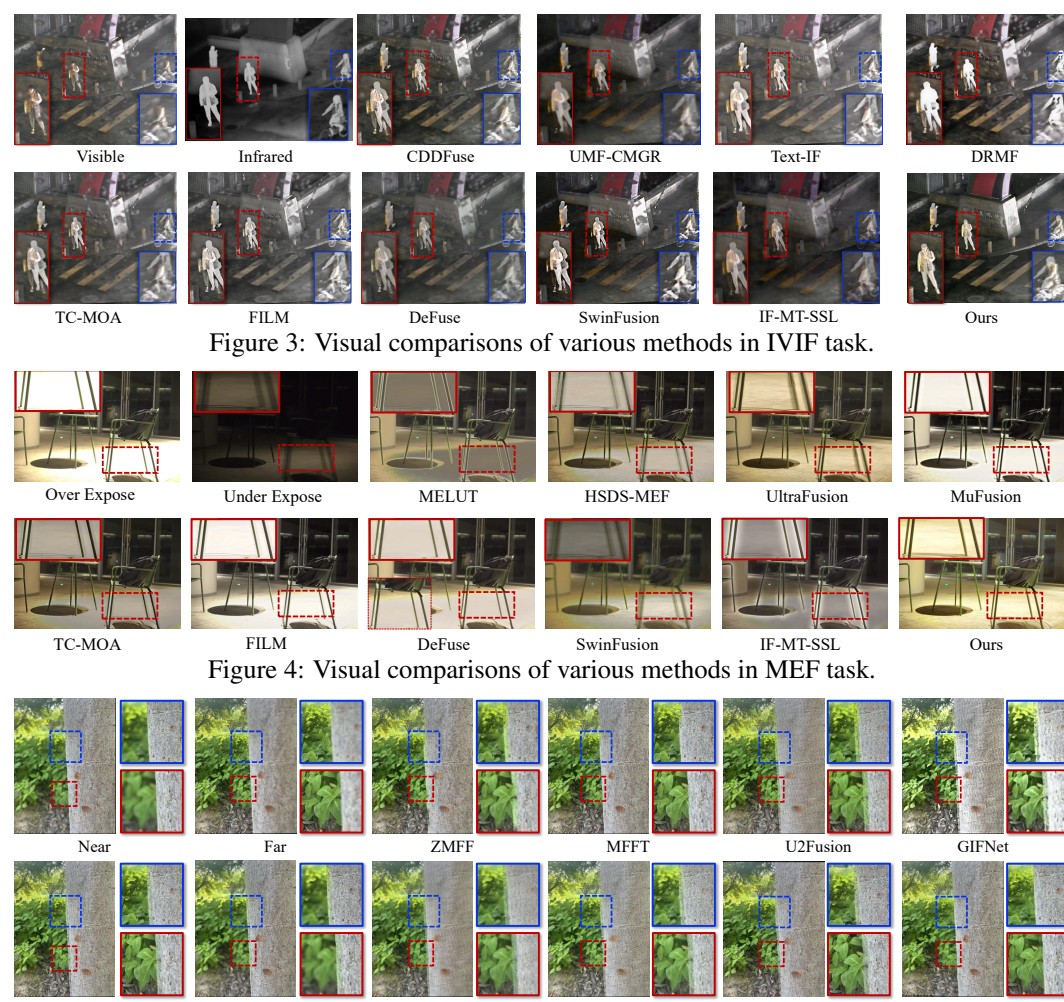

Figure 3: Visual comparisons of various methods in IVIF task.

Figure 4: Visual comparisons of various methods in MEF task.

Figure 5: Visual comparisons of various methods in MFF task.

organized around three key fusion tasks: (i) Infrared-Visible Fusion (IVIF). Scenarios are drawn from sources like LLVIP (Jia et al., 2021) and anti-UAV (Zhu et al., 2023), selected to cover a range of conditions from well-aligned inputs to challenging cases exhibiting inherent or simulated misalignment, dynamic objects, and adverse weather. (ii) Multi-Exposure Fusion (MEF). Scenarios are based on datasets like RealHDR-V (Shu et al., 2024) and DMEF (Tan et al., 2023), which feature prominent motion and ghosting artifacts. These challenges are often compounded with other common degradations such as blur and poor visibility. (iii) Multi-Focus Fusion (MFF). Scenarios are built upon datasets like RealMFF (Zhang et al., 2020) and EDMF (Li et al., 2024), presenting a diversity of challenges from simple focus stacking to complex cases involving dynamic objects, significant illumination variations, and adverse weather. The final dataset comprises 43,000 training pairs (22k IVIF, 8k MEF, 13k MFF) and a test set of 3,200 pairs (1.5k IVIF, 0.8k MEF, 1k MFF).

**Implementation details.** We adopt Qwen2.5-VL-7B-Instruct (Bai et al., 2025b) as the base model for FusionAgent. The supervised fine-tuning is performed on our 43,000 collected training samples, with a batch size of 2, a learning rate of 1e-5, and training for 2 epochs using the Llama-Factory framework (Zheng et al., 2024) on 8 A100 (80G) GPUs. The reinforcement learning, employing the GRPO-F algorithm, is conducted on a diverse subset of 5,000 challenging samples. For each step, we sample a batch of 4, a learning rate of 1e-6, and generate 4 responses per query, training for 2 epochs on 8 A100 (80G) GPUs. The detail hyper-parameter settings for SFT and GRPO-F are provided in Appendix C.1.

**Evaluation metrics.** We deviate from traditional fusion metrics (e.g., $Q_{abf}$, $Q_{cv}$, VIF), whose formulations are predicated on idealized assumptions of source image quality and alignment. Such assumptions are frequently violated by real-world degradations, causing the metric scores to become

Table 1: Comparison of decision-making strategies on multi-degraded fusion tasks. We evaluate different processing strategies across IVIF, MEF and MFF task. The averaged metrics presented are based on the common ones. **Best** results are highlighted.

| Strategy | IVIF Task | | | | | MEF Task | | | | |
|---|---|---|---|---|---|---|---|---|---|---|
| | MS-SSIM↑ | HyperIQA↑ | BRISQUE↓ | MUSIQ↑ | CLIPIQA↑ | MEF-SSIM$_d$↑ | HyperIQA↑ | BRISQUE↓ | MUSIQ↑ | CLIPIQA↑ |
| Random Order & Model | 0.2815 | 0.2522 | 48.3418 | 41.8833 | 0.2115 | 0.4533 | 0.3150 | 46.1824 | 43.1561 | 0.2898 |
| Random Order & Predict Model | 0.3582 | 0.3340 | 41.5227 | 48.2319 | 0.2640 | 0.5218 | 0.4011 | 40.5299 | 49.3782 | 0.3351 |
| Predict Order & Random Model | 0.4103 | 0.4281 | 33.1892 | 53.1524 | 0.3088 | 0.6304 | 0.5129 | 36.4183 | 56.8813 | 0.4529 |
| Pre-defined Order & Model | 0.4855 | 0.5891 | 19.5314 | 61.0581 | 0.3804 | 0.7782 | 0.6721 | 29.8371 | 68.1340 | 0.6215 |
| Zero-shot VLM Planner | 0.4629 | 0.5466 | 24.8816 | 58.7342 | 0.3571 | 0.7501 | 0.6503 | 31.7240 | 65.2199 | 0.5988 |
| Human Expert | 0.4988 | 0.6215 | 20.0253 | 63.5028 | 0.3819 | 0.7850 | 0.7018 | 27.9934 | 67.5028 | 0.6053 |
| **FuseAgent (Ours)** | **0.5064** | **0.6311** | **16.6587** | **64.3891** | **0.4122** | **0.8068** | **0.7105** | **27.2134** | **72.3164** | **0.6744** |

| Strategy | MFF Task | | | | | Average | | | |
|---|---|---|---|---|---|---|---|---|---|
| | MS-SSIM↑ | HyperIQA↑ | BRISQUE↓ | MUSIQ↑ | CLIPIQA↑ | HyperIQA↑ | BRISQUE↓ | MUSIQ↑ | CLIPIQA↑ |
| Random Order & Model | 0.5934 | 0.2811 | 49.5220 | 36.7291 | 0.2642 | 0.2828 | 48.0154 | 40.5895 | 0.2552 |
| Random Order & Predict Model | 0.6841 | 0.3205 | 44.8130 | 40.1824 | 0.3017 | 0.3519 | 42.2885 | 45.9308 | 0.3003 |
| Predict Order & Random Model | 0.7725 | 0.4593 | 38.2917 | 51.3888 | 0.3478 | 0.4668 | 35.9664 | 53.8075 | 0.3698 |
| Pre-defined Order & Model | 0.8870 | 0.6533 | 24.1821 | 64.9210 | 0.6401 | 0.6382 | 24.5169 | 64.7044 | 0.5473 |
| Zero-shot VLM Planner | 0.8653 | 0.6120 | 28.5294 | 62.0015 | 0.5905 | 0.6030 | 28.3783 | 61.9852 | 0.5155 |
| Human Expert | 0.8996 | 0.6686 | 22.3117 | 67.1305 | 0.6791 | 0.6640 | 23.4435 | 66.0718 | 0.5554 |
| **FuseAgent (Ours)** | **0.9051** | **0.6890** | **21.7826** | **68.0078** | **0.6870** | **0.6769** | **21.8849** | **68.2378** | **0.5912** |

decoupled from perceptual reality. Our quantitative assessment therefore first utilizes a suite of no-reference metrics robust to diverse degradations: HyperIQA (Su et al., 2020), BRISQUE (Mittal et al., 2012), MUSIQ (Ke et al., 2021) and CLIPIQA (Wang et al., 2023) to assess overall perceptual quality and semantic fidelity. Moreover, we employ specialized metrics for task-specific structural fidelity: (i) For IVIF, we use Spatial Frequency (SF) to measure information richness and the multi-scale MS-SSIM (Wang et al., 2003) to preserve multi-scale structural similarity (ii) For MFF, the multi-scale MS-SSIM is applied. (iii) For MEF involving motion, we adopt the dynamic-aware MEF-SSIMd (Fang et al., 2020).

**Tool settings.** FusionAgent's planned workflows are executed through a versatile suite of expert tools. Our implementation includes tools for registration (e.g., LoFTR (Sun et al., 2021)), optical flow estimation (e.g., RAFT (Teed & Deng, 2020)), image inpainting (e.g., LaMa (Suvorov et al., 2022)), illumination adjustment (e.g., IAT (Cui et al., 2022)), and image fusion (e.g., CDDFuse (Zhao et al., 2023)). Notably, we select representative and efficient models instead of the latest state-of-the-art models to simplify the validation process of our proposed paradigm. Incorporating more advanced models could further enhance performance. Refer to Appendix C.2 for a detailed description of the model settings.

## 4.2 DECISION MAKING CAPABILITY

**Compared methods.** To analyze the planning and decision-making capabilities of FusionAgent, we conducted a comparative study against a range of baseline strategies: (i) Random Order & Model, where both the workflow sequence and the expert model for each step are selected randomly. (ii) Random Order & Predicted Model, where the workflow order is random, but the model for each step is chosen by FusionAgent. (iii) Predicted Order & Random Model, where the workflow order is determined by FusionAgent, but the expert model for each step is chosen randomly. (iv) Pre-defined Order & Model, which represents a rigid, rule-based pipeline (e.g., restore → register → fuse) with a fixed set of models. (v) Zero-shot VLM Planner, where we use the base VLM (ChatGPT, Gemini) without any fine-tuning to generate the workflow. (vi) Human Expert, where an experienced specialist manually designs a bespoke workflow for each image. This baseline represents a strong, practical upper bound based on human intelligence.

**Results.** As shown in Table 1, we compare the decision-making capabilities of FuseAgent with baseline strategies. Three key observations emerge: 1) Random strategies exhibit significantly poor performance, highlighting the need for both strategic planning and proper model selection. 2) The "Predict Order & Random Model" strategy outperforms the "Random Order & Predict Model" strategy, suggesting that the optimal workflow sequence is more critical than the quality of individual tools. 3) Rule-based and expert-driven strategies, such as the Pre-defined pipeline (Avg. Hyper-IQA 0.6382) and the Human Expert (Avg. HyperIQA 0.6640), yield strong scores. However, our

Table 2: Quantitative comparison with state-of-the-art *task-specific methods* on multi-degraded fusion tasks. We evaluate across IVIF, MEF, and MFF benchmarks. We highlight the best and second-best results.

| Method | Task type | MS-SSIM↑ | HyperIQA↑ | MUSIQ↑ | CLIPIQA↑ | SF↑ |
|---|---|---|---|---|---|---|
| TarDAL | IVIF | 0.4636 | 0.4237 | 51.3410 | 0.2067 | 7.3909 |
| CDDFuse | IVIF | 0.4786 | 0.4800 | 57.1356 | 0.2789 | 7.2951 |
| C-MPDR | IVIF | 0.4928 | 0.2812 | 37.5873 | 0.1251 | 4.9812 |
| UMF-CMGR | IVIF | 0.4953 | 0.2844 | 36.7453 | 0.1310 | 4.6752 |
| Text-IF | IVIF | 0.3834 | 0.4739 | 57.3989 | 0.2715 | 8.2082 |
| DRMF | IVIF | 0.4851 | 0.4432 | 52.7181 | 0.3018 | 6.8013 |
| **FuseAgent (Ours)** | IVIF | 0.5064 | 0.6311 | 64.3891 | 0.4122 | 8.4295 |

| | | MS-SSIM↑ | HyperIQA↑ | MUSIQ↑ | CLIPIQA↑ | BRISQUE↓ |
|---|---|---|---|---|---|---|
| ZMFF | MFF | 0.7942 | 0.3410 | 50.2858 | 0.4062 | 41.2423 |
| MFFT | MFF | 0.8192 | 0.4222 | 52.0305 | 0.3293 | 32.2150 |
| MDLSR-RFM | MFF | 0.8107 | 0.4997 | 59.5366 | 0.4113 | 27.3327 |
| **FuseAgent (Ours)** | MFF | 0.9051 | 0.6890 | 68.0078 | 0.6870 | 21.7826 |

| | | MEF-SSIM$_d$↑ | HyperIQA↑ | MUSIQ↑ | CLIPIQA↑ | BRISQUE↓ |
|---|---|---|---|---|---|---|
| MEFLUT | MEF | 0.6970 | 0.5823 | 68.4800 | 0.4411 | 23.8959 |
| HSDS-MEF | MEF | 0.6228 | 0.5364 | 64.7659 | 0.3916 | 25.3278 |
| UltraFusion | MEF | 0.7740 | 0.5910 | 68.9866 | 0.4303 | 28.8180 |
| **FuseAgent (Ours)** | MEF | 0.8068 | 0.7105 | 72.3164 | 0.6744 | 25.2134 |

**FuseAgent** achieves the highest performance (Avg. HyperIQA **0.6769**), surpassing all baseline approaches. In conclusion, FuseAgent, augmented with the SFT+GRPO paradigm, discovers more optimal and generalizable strategies than static pipelines or case-by-case human heuristics for complex, real-world fusion problems. Detailed case studies that visualize this complete reasoning and planning process can be found in Appendix D.

### 4.3 GENERAL FUSION ABILITY

**Compared Methods.** To demonstrate the effectiveness and superiority of FuseAgent, we conduct comprehensive comparisons against a wide range of state-of-the-art (SOTA) methods, which can be broadly divided into two categories. The first category, **task-specific models**, includes methods designed for a single fusion task or a particular degradation, such as TarDAL (Liu et al., 2022), CDDFuse (Zhao et al., 2023), C-MPDR (Wang et al., 2024a), UMF-CMGR (Wang et al., 2022), Text-IF (Yi et al., 2024), DRMF (Tang et al., 2024), MEFLUT (Jiang et al., 2023), HSDS-MEF (Wu et al., 2024), UltraFusion (Chen et al., 2025), MDLSR-RFM(Wang et al., 2024c), ZMFF (Hu et al., 2023), and MFFT (Zhai et al., 2024). The second category consists of **all-in-one methods**, which aim to handle multiple fusion tasks within a single framework. This group includes U2Fusion (Xu et al., 2020), DeFuse (Liang et al., 2022), SwinFusion (Ma et al., 2022), IF-MT-SSL (Wang et al., 2024d), GIFNet (Cheng et al., 2025), MUFusion (Cheng et al., 2023), TC-MOA (Zhu et al., 2024b), and FILM (Zhao et al., 2024). For all compared methods, we use their officially released codes and follow the recommended settings to ensure a fair and rigorous comparison.

**Results.** As shown in Tables 2 and 3, FuseAgent consistently outperforms both task-specific and all-in-one models across our challenging in-the-wild benchmarks. While competing methods struggle with compounded defects, FuseAgent establishes a new state-of-the-art. For instance, in the IVIF task, it achieves a **36.6%** improvement in CLIPIQA score over the strongest specialized competitor, DRMF. Similarly, its performance in MEF (MEF-SSIM$_d$: **0.8068** vs. 0.7740) and MFF (HyperIQA: **0.6890** vs. 0.4946) surpasses the top competitors in each category. The visual comparisons in Figures 3-5 further confirm this superiority; unlike static methods that produce artifacts like ghosting and distortion, FuseAgent's dynamic, perception-driven planning yields clean and structurally coherent results.

### 4.4 ABLATION STUDY

**Training Strategy.** To validate our two-stage training paradigm, we compare three configurations in Table 4 (rows 2–4). The SFT-only approach establishes a reasonable baseline but is insufficient for optimizing complex scenarios that deviate from the expert trajectories. Conversely, training with

Table 3: Quantitative comparison with state-of-the-art ***all-in-one methods*** on multi-degraded fusion tasks. We evaluate across IVIF, MEF, and MFF benchmarks. We highlight the best and second-best results.

| Method | IVIF | | | | | MEF Task | | | | MFF Task | | | |
|---|---|---|---|---|---|---|---|---|---|---|---|---|---|
| | SF↑ | MS-SSIM↑ | HyperIQA↑ | MUSIQ↑ | CLIPIQA↑ | MEF-SSIM$_d$↑ | HyperIQA↑ | MUSIQ↑ | CLIPIQA↑ | MS-SSIM↑ | HyperIQA↑ | MUSIQ↑ | CLIPIQA↑ |
| U2Fusion | 6.9949 | 0.5054 | 0.3948 | 53.5210 | 0.2771 | 0.6768 | 0.4782 | 62.2200 | 0.4246 | 0.8560 | 0.4078 | 53.7902 | 0.4408 |
| DeFuse | 5.7597 | 0.4757 | 0.4720 | 54.7675 | 0.2388 | 0.7461 | 0.4893 | 59.7320 | 0.3972 | 0.8813 | 0.3981 | 49.9259 | 0.3668 |
| SwinFusion | 7.3620 | 0.4764 | 0.4049 | 51.8339 | 0.2646 | 0.6129 | 0.4170 | 55.5563 | 0.3208 | 0.8413 | 0.3614 | 52.6268 | 0.3283 |
| IF-MT-SSL | 6.2778 | 0.5042 | 0.4521 | 51.9412 | 0.2529 | 0.6863 | 0.5894 | 66.9227 | 0.4979 | 0.8372 | 0.4946 | 59.5262 | 0.4501 |
| GIFNet | 7.6257 | 0.4996 | 0.3956 | 55.1952 | 0.2760 | 0.5483 | 0.5312 | 63.6882 | 0.4414 | 0.7783 | 0.4414 | 58.3438 | 0.4049 |
| MUFusion | 7.3898 | 0.4372 | 0.2572 | 41.4161 | 0.1361 | 0.6852 | 0.3997 | 51.0261 | 0.3182 | 0.8170 | 0.4243 | 58.2549 | 0.4629 |
| TC-MOA | 6.3922 | 0.4753 | 0.4010 | 52.2921 | 0.2289 | 0.7041 | 0.5744 | 66.8931 | 0.4775 | 0.8289 | 0.4228 | 53.8276 | 0.3914 |
| FILM | 7.6726 | 0.4603 | 0.5136 | 59.5164 | 0.2956 | 0.7518 | 0.5763 | 67.3389 | 0.5084 | 0.8322 | 0.4935 | 57.1397 | 0.4210 |
| **FuseAgent (Ours)** | 8.4295 | 0.5064 | 0.6311 | 64.3891 | 0.4122 | 0.8068 | 0.7105 | 72.3164 | 0.6744 | 0.9051 | 0.6890 | 68.0078 | 0.6870 |

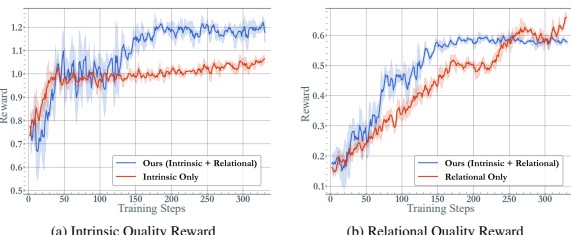

(a) Intrinsic Quality Reward     (b) Relational Quality Reward

Figure 6: The reward trends across training steps.

Table 4: Ablation studies on different training strategies and reward design.

| Configurations | SF↑ | HyperIQA↑ | BRISQUE↓ | CLIPIQA↑ |
|---|---|---|---|---|
| Training strategy | | | | |
| only SFT | 8.10 | 0.45 | 25.43 | 0.39 |
| only RL | 7.79 | 0.55 | 25.88 | 0.44 |
| SFT + RL (Ours) | **8.63** | **0.68** | **21.88** | **0.59** |
| Reward design | | | | |
| only IQR | 8.45 | 0.61 | 23.02 | 0.54 |
| only RQR | **8.81** | 0.52 | 24.15 | 0.4 |
| IQR + RQR (Ours) | 8.63 | **0.68** | **21.88** | **0.59** |

GRPO-F from scratch improves perceptual metrics at the cost of structural fidelity. This trade-off indicates that while RL can explore better perceptual solutions, it struggles to acquire foundational fusion knowledge without proper initialization. The proposed SFT+GRPO-F pipeline attains superior performance across all metrics, confirming that SFT provides a crucial policy initialization, creating a robust foundation upon which GRPO-F effectively refines the policy for real-world complexities without sacrificing structural coherence.

**Reward Design.** The efficacy of our multi-dimensional reward is analyzed in Table 4 (rows 6–8) and Figure 6. Isolating the relational quality reward ($R_{\text{relational}}$) yields the highest structural fidelity, confirming its role in optimizing for geometric and structural consistency, but provides insufficient signal for improving standalone perceptual quality. Conversely, optimizing for only the intrinsic quality reward ($R_{\text{intrinsic}}$) leads to premature convergence; as illustrated in Figure 6(a), the policy rapidly improves but plateaus at a suboptimal level after approximately 50 training steps, indicating an inability to escape local optima. The full reward combination ($R_{\text{intrinsic}} + R_{\text{relational}}$) achieves the best overall performance, excelling in both perceptual quality and artifact reduction. This synergy validates our design: the intrinsic term ensures perceptual quality, while the relational term provides the necessary guidance to escape local optima and enforce inter-image compatibility, leading to solutions that are both visually pleasing and structurally coherent.

## 5 CONCLUSION

In this paper, we present FuseAgent, a VLM-powered agent system designed to address the challenges of in-the-wild image fusion. FuseAgent autonomously perceives complex, coupled degradations and dynamically orchestrates expert models into tailored workflows. The training process follows a two-stage optimization paradigm: 1) Supervised Fine-Tuning (SFT) on a curated dataset of expert trajectories instills foundational skills in degradation perception and tool usage. 2) Group Relative Policy Optimization for Fusion (GRPO-F), augmented with novel, unsupervised rewards—the Intrinsic Quality Reward (IQR) and Relational Quality Reward (RQR)—refines the agent's decision-making and generalization abilities. Comprehensive experiments on challenging in-the-wild benchmarks demonstrate that FuseAgent significantly outperforms both specialized and all-in-one models in fusion quality and decision-making.

ETHICS STATEMENT

This work complies with the ICLR Code of Ethics. We confirm that all authors have read and agreed to abide by its principles. Our research does not involve human subjects or sensitive personal data, and no potentially harmful insights are presented. The datasets used are publicly available and appropriately cited. We have no known conflicts of interest, financial or otherwise, related to this work. All methodologies were designed with fairness and transparency in mind, and we have taken steps to ensure reproducibility and research integrity. No IRB approval was required for this study.

REPRODUCIBILITY STATEMENT

We are committed to ensuring the reproducibility of our research. In the main paper, we provide a detailed description of the model's architecture, experimental setup, and evaluation methods. To provide deep insight into the model's internal operational mechanisms, we have included an interactive visualization interface in the supplementary materials. This interface dynamically presents the model's end-to-end inference process, from receiving the initial input to generating the final output, thereby revealing its complex decision-making logic.

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

APPENDIX

Our Appendices includes the following sections:

## USE OF LLMs

We used a large language model (LLM) solely for proofreading purposes, such as correcting grammatical errors and improving sentence clarity in the final draft. The LLM did not contribute to the research ideation, experimental design, theoretical analysis, or any substantive content of this work. All scientific content, claims, and results are the sole responsibility of us.

## A    ADDITIONAL METHOD DETAILS

### A.1    GROUP RELATIVE POLICY OPTIMIZATION

Our reinforcement learning stage employs Group Relative Policy Optimization (GRPO) (Shao et al., 2024), a policy-gradient algorithm that operates without a critic model. In GRPO, given an input image pair, the policy model $\pi_\theta$ generates a set of $N$ potential action trajectories $\{\mathcal{A}_1, \mathcal{A}_2, \ldots, \mathcal{A}_N\}$. Each trajectory is executed, and a final reward is computed for each, resulting in a set of rewards $\{R_1, R_2, \ldots, R_N\}$. Unlike PPO, which relies on a value function, GRPO directly compares the rewards within the group to determine the relative quality of each trajectory. The relative advantage $A_i$ of the i-th trajectory is calculated by normalizing its reward against the statistics of the entire group:

$$A_i = \frac{R_i - \text{Mean}(\{R_1, R_2, \ldots, R_N\})}{\text{Std}(\{R_1, R_2, \ldots, R_N\}) + \epsilon},$$

where Mean and Std represent the mean and standard deviation of the rewards, and $\epsilon$ is a small constant for numerical stability. This normalization allows GRPO to capture nuanced differences between candidate trajectories. The policy update is constrained by minimizing the KL divergence between the current and reference models, ensuring stable learning. For more details, we refer the reader to (Guo et al., 2025; Jaech et al., 2024).

### A.2    DETAILS OF REWARD CALCULATION

This section provides the detailed formulations for the two primary reward components introduced in Sec. 3.3.2: the Intrinsic Quality Reward and the Relational Quality Reward.

- **Intrinsic Quality Reward** ($R_{\textbf{intrinsic}}$). This reward assesses the standalone perceptual quality of a single image $I'_t$ resulting from a unilateral action (e.g., restoration, initial enhancement). It encourages artifact removal and visual plausibility. The reward is a weighted

composite score from a suite of no-reference Image Quality Assessment (IQA) metrics, each chosen to capture different aspects of quality:

$$R_{\text{intrinsic}}(I'_t) = \sum_{m \in M_{\text{IQA}}} w_m \cdot \hat{\Phi}_m(I'_t),$$

where $M_{\text{IQA}}$ = {HyperIQA, BRISQUE, MUSIQ, CLIPIQA}. For instance, BRISQUE is sensitive to common compression and sensor artifacts, MUSIQ assesses overall aesthetic quality, and CLIPIQA provides a semantic evaluation of fidelity.

- **Relational Quality Reward** ($R_{\text{relational}}$). This reward is the core of our RQA paradigm and evaluates any action that modifies or depends on the relationship between the two images in a pair. It is a multi-dimensional assessment whose specific instantiation depends on the action's goal.

  **Enhancement Compatibility.** For actions that enhance one image (e.g., visible light enhancement producing $I'_v$) to better match another (e.g., an infrared image $I_i$), the reward measures the improvement in fusion compatibility. It is a weighted sum over several dimensions:

  $$R_{\text{enhance}}(I'_v, I_v, I_i) = \sum_{d \in D_{\text{enhance}}} w_d \cdot \mathcal{C}_d(I'_v, I_v, I_i),$$

  where key dimensions $d \in D_{\text{enhance}}$ include *structural compatibility* (via SSIM), *gradient compatibility* (via gradient correlation), and *contrast compatibility* (via histogram distance), ensuring the enhancement prepares the image for a more effective final fusion.

  **Alignment and Deghosting Accuracy.** For actions that address geometric or temporal misalignments (e.g., registration, deghosting via optical flow and masking), the reward assesses the resulting consistency. As ground-truth transformations are unavailable, the reward is a composite score evaluating alignment from multiple perspectives:

  $$R_{\text{align}}(I'_{\text{warped}}, I_{\text{ref}}) = \sum_{d \in D_{\text{align}}} w_d \cdot \mathcal{A}_d(I'_{\text{warped}}, I_{\text{ref}}),$$

  where dimensions $d \in D_{\text{align}}$ include *structural consistency* (measured by NCC), *feature similarity* (cosine similarity of deep features from a pre-trained VGG network), and *geometric precision* (e.g., penalizing unrealistic distortions).

  **Overall Fusion Quality.** For the final fusion action which produces the output $I_f$, this reward serves as the comprehensive evaluation of the entire workflow. It holistically measures how well $I_f$ integrates information from both sources, $I_v$ and $I_i$. It combines reference-based metrics that assess structural and informational fidelity with no-reference metrics for perceptual quality:

  $$R_{\text{fusion}}(I_f, I_v, I_i) = \sum_{m \in M_{\text{fusion}}} w_m \cdot \Phi_m(I_f, I_v, I_i),$$

  where $M_{\text{fusion}}$ includes information-theoretic metrics (SF, AG), structural metrics (MS-SSIM), and the full suite of IQA metrics.

## B  DETAILS OF DATASET

This section provides a detailed account of the three-stage pipeline used to construct the training and evaluation dataset for FuseAgent, as introduced in Sec. 3.2. Our goal was to create a large-scale, diverse dataset of complex fusion scenarios paired with high-quality expert reasoning (Chain-of-Thought, CoT) and optimal action trajectories.

### B.1  STAGE I: IMAGE PAIR GENERATION AND DEGRADATION SYNTHESIS

Our data generation process employs a hybrid strategy, combining authentic degraded image pairs from public benchmarks with systematically synthesized data to ensure both realism and comprehensive coverage of challenges.

**Image Collection.** We source our data from a variety of public datasets covering our three main fusion tasks (IVIF, MEF, MFF), including real-world captures that already exhibit a single, specific degradation (e.g., misalignment from anti-UAV, dynamic artifacts from RealHDR-V). These serve as the foundation for our benchmark.

**Degradation Synthesis.** To create complex, compounded challenges that are underrepresented in existing datasets, we developed a modular degradation library. This library is applied to both clean source images and real degraded images to layer additional defects. Key simulated degradations include:

- **Geometric Misalignment:** We apply random homography and affine transformations to one image in a pair to simulate camera shake and parallax errors.
- **Dynamic Artifacts:** To simulate motion, we first segment a chosen object, inpaint the background using a high-fidelity inpainting model Suvorov et al. (2022), and then re-insert the object with a slight transformation in one of the images.
- **Photometric Inconsistencies:** We model exposure and white balance variations by applying randomized color and brightness adjustments based on physical camera response functions.
- **Adverse Weather & Sensor Noise:** We utilize established physical models and generative techniques to synthesize realistic weather effects (e.g., rain, haze) and sensor noise patterns.

This hybrid and modular approach allows us to generate a rich dataset spanning a wide spectrum of complexity, from simple single-defect cases to challenging scenarios with multiple, coupled degradations.

### B.2 Stage II: Expert Response Generation

For each degraded image pair, we generate a corresponding expert response, which consists of the optimal action trajectory and the underlying CoT rationale.

**Optimal Action Trajectory.** To establish a ground-truth sequence of tool-calls ($\mathcal{A}$), we employ an exhaustive search strategy. For each scenario, we define a set of applicable pre-processing and fusion tools from our library. We then explore all valid permutations of tool sequences, executing each full trajectory. Each final fused image is scored using our Holistic Quality Assessment (HQA) reward function (detailed in Appendix A.2). The trajectory that yields the highest HQA score is designated as the ground-truth optimal path.

**Chain-of-Thought (CoT) Rationale.** To generate a human-like reasoning process ($\mathcal{C}$) that explains the logic behind the optimal trajectory, we "reverse-engineer" the thought process using a powerful VLM. We provide the VLM with the degraded source images and the pre-determined optimal action trajectory. The VLM is then prompted to produce a concise, first-person rationale explaining *why* this sequence of actions is necessary and logical. This process ensures that the CoT is not only coherent but also perfectly aligned with the ground-truth actions. The detailed prompt used for this generation process is presented in Table 8.

## C Additional Experiment Details

### C.1 Hyper-parameter Settings

In Table 5, we detail the hyper-parameter settings for our experiments.

### C.2 Tool settings

Table 6 lists the comprehensive suite of tools integrated into our framework. For registration and alignment, we utilize both a detector-free model, LoFTR Sun et al. (2021), and the classic SIFT Lowe (1999) algorithm as a robust baseline. Motion analysis is handled by GMA Jiang et al. (2021) and the iterative RAFT Teed & Deng (2020) for accurate optical flow estimation. For inpainting and filling masked regions, we employ LaMa Suvorov et al. (2021) and Inpaint-Anything Yu et al. (2023) for high-resolution inpainting.

Table 5: Hyper-parameter settings for SFT and GRPO-F

| Hyper-parameter | SFT | GRPO-F |
|---|---|---|
| Batch size | 2 | 4 |
| Learning rate | 1e-5 | 1e-6 |
| Weight decay | 0 | 0 |
| Optimizer | AdamW | AdamW |
| Warmup ratio | 0.1 | 0.1 |
| LR scheduler | cosine | cosine |
| Training samples | 43K | 5K |
| Training epochs | 2 | 2 |
| Precision | bfloat16 | bfloat16 |
| KL coefficient | - | 0.1 |
| Reward coefficients | - | $IQR$:1, $RQR$:1 |
| Number of generations | - | 4 |
| GPU resources | 8×A100 (∼384 GPU hours) | 8×A100 (∼2076 GPU hours) |

Our denoising module consists of the hybrid SCUnet Zhang et al. (2023) and the efficient transformer-based Restormer Zamir et al. (2022). To address adverse weather conditions, we use IDT Xiao et al. (2022) for deraining and RIDCP Wu et al. (2023b) for dehazing. A versatile set of tools is available for general restoration tasks like deblurring and super-resolution, including the diffusion-based StableSR-turbo Wang et al. (2024b) and the fast GAN-based Real-ESRGAN Wang et al..

For challenges in illumination and exposure, our framework is equipped with a wide array of methods: a fast low-light enhancer in Img2img-turbo-night Parmar et al. (2024), a lightweight transformer IAT Cui et al. (2022), a diffusion-based model LightenDiff Jiang et al. (2024), and two classical histogram-based techniques, Histogram Matching and CLAHE Reza (2004).

Finally, the critical task of merging processed outputs is handled by a diverse set of fusion operators. In addition to the versatile SwinFusion Ma et al. (2022), our suite includes models for various scenarios: Text-IF Yi et al. (2024) for text-guided interactive fusion, CDDFuse Zhao et al. (2023) for multi-modal tasks, MEFLUT Jiang et al. (2023) for efficient multi-exposure fusion, as well as MFFT Zhai et al. (2024) and the zero-shot ZMFF Hu et al. (2023) for multi-focus fusion.

Notably, some models lack weights corresponding to certain tasks but are inherently adaptable; we collect appropriate data to retrain them. It is also important to note that we are not necessarily utilizing the latest state-of-the-art tools, suggesting considerable potential for future enhancements to our models.

## C.3 ZERO-SHOT VLM PLANNER BASELINE

To evaluate the contribution of our two-stage training paradigm (SFT+GRPO-F), we established a strong baseline using a zero-shot VLM planner. This baseline utilizes the base VLM (e.g., Qwen2.5-VL-7B-Instruct, GPT 4V) without any of the domain-specific fine-tuning described in our main paper.

The agent's behavior in this setting is guided solely by a comprehensive textual prompt that instructs it to act as an image fusion expert. This prompt, detailed in Table 8, provides the model with the high-level task objective, the required output format (including the use of `<think>` and `<answer>` tags), and the complete library of available expert tools and their functions. The VLM is then tasked with generating a full action trajectory based on its general-purpose, pre-existing reasoning capabilities, without any in-domain examples or specialized training. This allows us to fairly measure the performance gains achieved through our targeted SFT and GRPO-F stages.

Table 6: The expert tool suite available to FuseAgent, covering a wide range of pre-processing and fusion operations.

| Task | Tools | Model Description |
|---|---|---|
| **Registration & Alignment** | LoFTR | Detector-free local feature matching model that excels at finding correspondences in challenging conditions. |
| | SIFT | Classic Scale-Invariant Feature Transform algorithm, serving as a robust baseline for keypoint matching. |
| **Optical Flow Estimation** | GMA | A global motion aggregation-based network for accurate optical flow estimation, crucial for motion analysis in dynamic scenes. |
| | RAFT | An iterative deep learning model for optical flow that uses a recurrent GRU-based operator to refine predictions. |
| **Inpainting & De-occlusion** | LaMa | High-resolution inpainting model using fast Fourier convolutions, effective at filling large masked regions after deghosting. |
| | Inpaint-Anything | A mask-free inpainting framework based on SAM that enables users to remove, fill, or replace objects via simple clicks. |
| **Denoising** | SCUnet | Hybrid UNet-based model combining convolution and transformer blocks for robust real-world denoising. |
| | Restormer | Efficient Transformer-based model for high-quality image denoising and restoration. |
| **Deraining & Dehazing** | IDT | Transformer-based model for unified de-raining and raindrop removal. |
| | RIDCP | Efficient dehazing model utilizing high-quality codebook priors for complex real-world haze. |
| **Deblurring & Super-resolution** | StableSR-turbo | Utilizes pre-trained diffusion models for high-quality super-resolution, deblurring, and artifact removal. |
| | Real-ESRGAN | Fast GAN-based model for super-resolution and deblurring, handling complex real-world degradations efficiently. |
| **Illumination & Exposure Correction** | Img2img-turbo-night | Fast and efficient model based on SD-turbo, designed for low-light enhancement. |
| | ITA | Lightweight transformer for efficient low-light and exposure correction. |
| | LightenDiff | Diffusion-based framework for unsupervised low-light enhancement leveraging Retinex theory. |
| | Histogram Matching | Classical technique that modifies an image's histogram to match a reference, used for standardizing exposure. |
| | CLAHE | Contrast Limited Adaptive Histogram Equalization, a robust method for enhancing local contrast in images. |
| **General-Purpose Fusion** | SwinFusion | A Swin Transformer-based model serving as a powerful and versatile final fusion operator. |
| | Text-IF | A novel fusion model that leverages semantic text guidance for degradation-aware and interactive fusion tasks. |
| | CDDFuse | A Correlation-Driven feature Decomposition Fusion network using a dual-branch Transformer-CNN to model cross-modality features. |
| | MEFLUT | An efficient Multi-Exposure Fusion method that encodes fusion weights into a 1D lookup table (LUT) using attention. |
| | MFFT | A Multi-Focus Fusion method using an interactive transformer and asymmetric soft sharing to produce all-in-focus images. |
| | ZMFF | A zero-shot, untrained Multi-Focus Fusion framework that models the deep prior of the image and focus map. |

## D  CASE STUDIES OF THE DECISION-MAKING PROCESS

This section provides two detailed case studies that visualize the complete reasoning and planning workflow of FuseAgent when confronted with complex, in-the-wild fusion scenarios. Each case study (Figure 7 and 8) is structured to offer a transparent view of the agent's process, from perception to execution. The visual workflow at the top of each figure illustrates the step-by-step application of the planned action trajectory, showing the intermediate images generated at each stage. Below the visuals, the agent's corresponding textual output reveals its Chain-of-Thought (CoT) reasoning within `<think>` tags, followed by the final, executable plan in `<answer>` tags.

*Action Details*

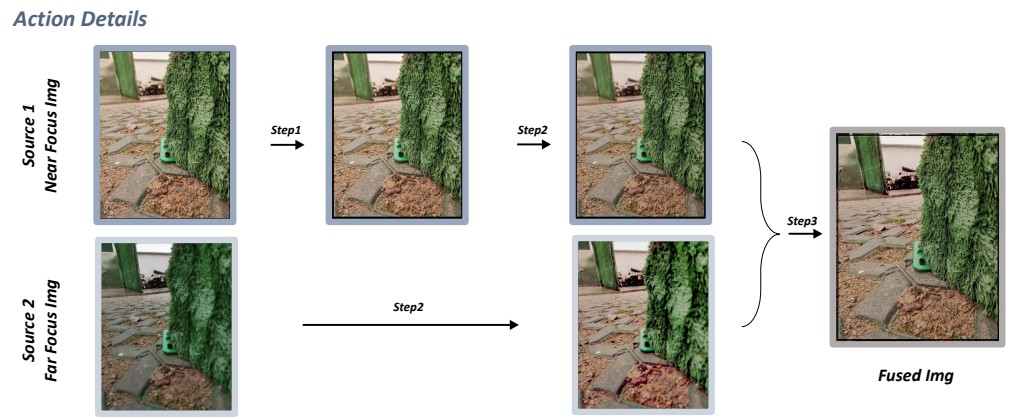

**FuseAgent:**

**\<think\>** This scene, depicting a person in a dark street with a bright streetlight, presents a challenging image fusion task. My initial analysis reveals several issues. First, the visible light image is severely degraded by low light, noise, and glare, which obscures key details and makes it difficult to extract reliable feature points. This prevents a straightforward registration process. Therefore, I must first perform restoration to enhance the image quality and prepare it for alignment. **\</think\>**

**\<answer\>** [task:deflare] [model:*flare7k*]➡[task:lighten] [model:*hvicident*]➡[task:denoise] [model:*scunet*] **\</answer\>**

**\<think\>** Given the lack of robust features in both the visible and infrared images, a traditional feature-based registration approach would likely fail. Instead, I'll need to use a more powerful model, such as a deep learning-based method, to accurately predict the dense deformation field. **\</think\>**

**\<answer\>**[task:registration] [model:*loftr*] **\</answer\>**

**\<think\>** Finally, while the visible light image is restored, it remains an under-exposed image. This requires the use of a specialized fusion model that is optimized for low-light conditions to effectively combine the textural information from the visible spectrum with the thermal properties captured by the infrared sensor. **\</think\>**

**\<answer\>**[[task:fusion] [model:*textif*] **\</answer\>**

Figure 7: FuseAgent's Strategy for a Complex In-the-Wild Scenario: Tackling Compounded Degradations in Nighttime Fusion.

*Action Details*

**FuseAgent:**

**\<think\>** The user wants to perform multi-focus image fusion. I've been given two images of a moss-covered object on a tiled ground, likely taken at different focus points. I see a slight misalignment between the two images, which is probably due to camera shake during capture. The first step is to perform pixel-level alignment by adjusting the angle of one of the images. **\</think\>**

**\<answer\>**[task:registration] [model:*sift*] **\</answer\>**

**\<think\>** Next, I'll address the differences in exposure and white balance caused by the camera's auto-adjustment. To prevent ghosting artifacts in the final fused image, I'll need to adjust the brightness and color balance of both images to ensure they are consistent. **\</think\>**

**\<answer\>**[task:expose correction] [model: *Histogram Matching*] **\</answer\>**

**\<think\>** After these preparatory steps, the images will meet the necessary conditions for a clean fusion. I can then proceed with the fusion process to create a single, all-in-focus image. **\</think\>**

**\<answer\>**[task:fusion] [model:*MFFT*] **\</answer\>**

Figure 8: FuseAgent's Strategy for a Complex In-the-Wild Scenario: Tackling Geometric and Photometric Inconsistencies in Multi-Focus Fusion.

Table 7: Prompt for Producing Refined CoT Annotations

---

### Prompt for Producing Refined CoT Annotations

⋯⋯⋯⋯⋯⋯⋯⋯⋯⋯⋯⋯⋯⋯⋯⋯⋯⋯⋯⋯⋯⋯⋯⋯⋯⋯⋯⋯⋯⋯⋯⋯⋯⋯⋯⋯⋯⋯⋯⋯⋯⋯⋯⋯⋯⋯⋯

As an image processing specialist, you're tasked with reverse-engineering the thought process behind an ¡Fusion Type¿. Given the source images, the pre-fusion processing pipeline, and the fusion algorithm, deduce the rationale for the entire workflow.

**Requirements:**
1. Your response should be a concise, single-paragraph analysis, written in English, limited to 400 characters.
2. Please omit any mention of specific method names, as they are part of the internal thought process and not required in the final response.
3. To start the analysis, provide **a brief introduction to the pre-fusion image**. For example, both images to be fused show a girl on the grass, playing with her dog.
4. For each processing operation, the introduction should **emphasize why it is necessary**. Present the rationale before describing how it is performed.
5. The final fused image should not be mentioned; it serves only as a reference to inform our speculation about the expert's original thinking.
6. The final output should sound like natural spoken English, not written text, and be from the original expert's perspective (first-person perspective).

**Notes:**
- The images, in order, are a visible light shot, an infrared shot, and the fused result.
- Common pre-fusion image processing methods:
    1. Registration & Alignment: (Method name: characteristics)
        - LoFTR: Detector-free local feature matching model that excels at finding correspondences in challenging conditions.
        - SIFT: Classic Scale-Invariant Feature Transform algorithm, serving as a robust baseline for keypoint matching.
        - …
    2. Optical Flow Estimation: (Method name: characteristics)
        - GMA: A global motion aggregation-based network for accurate optical flow estimation, crucial for motion analysis in dynamic scenes.
        - RAFT: An iterative deep learning model for optical flow that uses a recurrent GRU-based operator to refine predictions.
        - …
    3. Inpainting & De-occlusion: (Method name: characteristics)
        - LaMa: High-resolution inpainting model using fast Fourier convolutions, effective at filling large masked regions after deghosting.
        - Inpaint-Anything: A mask-free inpainting framework based on SAM that enables users to remove, fill, or replace objects via simple clicks.
        - …
    4. Denoising: (Method name: characteristics)
        - SCUnet: Hybrid UNet-based model combining convolution and transformer blocks for robust real-world denoising.
        - Restormer: Efficient Transformer-based model for high-quality image denoising and restoration.
        - …
    5. Deraining & Dehazing: (Method name: characteristics)
        - IDT: Transformer-based model for unified de-raining and raindrop removal.
        - RIDCP: Efficient dehazing model utilizing high-quality codebook priors for complex real-world haze.
        - …
    6. Deblurring & Super-resolution: (Method name: characteristics)
        - StableSR-turbo: Utilizes pre-trained diffusion models for high-quality super-resolution, deblurring, and artifact removal.
        - Real-ESRGAN: Fast GAN-based model for super-resolution and deblurring, handling complex real-world degradations efficiently.
        - …
    7. Illumination & Exposure Correction: (Method name: characteristics)
        - Img2img-turbo-night: Fast and efficient model based on SD-turbo, designed for low-light enhancement.
        - ITA: Lightweight transformer for efficient low-light and exposure correction.
        - LightenDiff: Diffusion-based framework for unsupervised low-light enhancement leveraging Retinex theory.
        - Histogram Matching: Classical technique that modifies an image's histogram to match a reference, used for standardizing exposure.
        - CLAHE: Contrast Limited Adaptive Histogram Equalization, a robust method for enhancing local contrast in images.
        - …
    8. General-Purpose Fusion: (Method name: characteristics)
        - SwinFusion: A Swin Transformer-based model serving as a **powerful and versatile final fusion** operator.
        - Text-IF: A novel fusion model that leverages semantic text guidance for **degradation-aware and interactive fusion tasks**.
        - CDDFuse: A Correlation-Driven feature Decomposition Fusion network using a dual-branch Transformer-CNN to **model cross-modality features**.
        - MEFLUT: An efficient **Multi-Exposure Fusion** method that encodes fusion weights into a 1D lookup table (LUT) using attention.
        - MFFT: A **Multi-Focus Fusion** method using an interactive transformer and asymmetric soft sharing to produce all-in-focus images.
        - ZMFF: A zero-shot, **untrained Multi-Focus Fusion** framework that **models the deep prior of the image and focus map**.
        - …
    …

The pre-fusion processing on the first image: ¡Method¿
The pre-fusion processing on the second image: ¡Method¿
Method for fusion: ¡Method¿

---

Table 8: The prompt used for the Zero-shot VLM Planner baseline.

---

### Prompt for Zero-Shot VLM Planner

---

As an expert in image fusion and processing, your task is to analyze a given pair of source images for an `<Fusion Type>` task. You must identify any degradations or inconsistencies and then generate an optimal, step-by-step workflow to produce a high-quality fused image using the provided tool library.

**Requirements:**

1. Your response must consist of two parts: a reasoning process and a final action plan.

2. First, provide your step-by-step analysis and reasoning within `<think>` tags. Explain the problems you see and why your proposed plan is the correct approach.

3. Second, provide the final, executable workflow as a sequence of tool-calls within `<answer>` tags. Each tool-call should be in the format [task:task_name][model:model_name].

4. You must only select tools and models from the Tool Library provided below. Do not invent new ones.

5. The order of operations is critical. Your plan should address necessary pre-processing steps (like registration or restoration) before the final fusion.

**Tool Library:**

- You will be provided with two source images for a specific fusion task.
- The available tools and their functions are listed below:
    1. Registration & Alignment: (Tool name: function)
        - LoFTR: Detector-free local feature matching model that excels at finding correspondences in challenging conditions.
        - SIFT: Classic Scale-Invariant Feature Transform algorithm, serving as a robust baseline for keypoint matching.
        - …
    2. Optical Flow Estimation: (Tool name: function)
        - GMA: A global motion aggregation-based network for accurate optical flow estimation, crucial for motion analysis in dynamic scenes.
        - RAFT: An iterative deep learning model for optical flow that uses a recurrent GRU-based operator to refine predictions.
        - …
    3. Denoising: (Tool name: function)
        - SCUnet: Hybrid UNet-based model combining convolution and transformer blocks for robust real-world denoising.
        - Restormer: Efficient Transformer-based model for high-quality image denoising and restoration.
        - …
    4. Deraining & Dehazing: (Tool name: function)
        - IDT: Transformer-based model for unified de-raining and raindrop removal.
        - RIDCP: Efficient dehazing model utilizing high-quality codebook priors for complex real-world haze.
        - …
    5. Deblurring Super-resolution: (Tool name: function)
        - StableSR-turbo: Utilizes pre-trained diffusion models for high-quality super-resolution, deblurring, and artifact removal.
        - Real-ESRGAN: Fast GAN-based model for super-resolution and deblurring, handling complex real-world degradations efficiently.
        - …
    6. Illumination Exposure Correction: (Tool name: function)
        - Img2img-turbo-night: Fast and efficient model based on SD-turbo, designed for low-light enhancement.
        - CLAHE: Contrast Limited Adaptive Histogram Equalization, a robust method for enhancing local contrast in images.
        - …
    …
    7. General-Purpose Fusion: (Tool name: function)
        - SwinFusion: A Swin Transformer-based model serving as a powerful and versatile final fusion operator.
        - Text-IF: A novel fusion model that leverages semantic text guidance for degradation-aware and interactive fusion tasks.
        - CDDFuse: A Correlation-Driven feature Decomposition Fusion network using a dual-branch Transformer-CNN.
        - …

