# OpenReview forum: "FuseAgent: A VLM-driven Agent for Unified In-the-Wild Image Fusion"
_ICLR.cc/2026/Conference — Submitted to ICLR 2026_

### Official Review · Reviewer_mtHR · 2025-10-24

**Soundness:** 3
**Presentation:** 3
**Contribution:** 3
**Rating:** 6
**Confidence:** 5

**Summary:**

This paper proposes FuseAgent, a VLM-driven agent that acts as a controller to dynamically plan and execute a sequence of expert models for "in-the-wild" image fusion. The agent is trained in two stages: first, using Supervised Fine-Tuning (SFT) on a curated dataset of expert trajectories , and second, using a reinforcement learning algorithm (GRPO-F) with a novel multi-dimensional reward signal on unlabeled data. The agentic framework itself and the SFT+RL training pipeline, including the Intrinsic and Relational Quality Rewards (IQR/RQR), are the key technical contributions.
The model demonstrates superior performance compared to existing task-specific and all-in-one fusion methods, and even outperforms manually designed "Human Expert" pipelines in quantitative evaluations. The paper also contributes a new data generation pipeline for creating complex, multi-degradation fusion scenarios and a comprehensive evaluation benchmark.

**Strengths:**

1. The primary strength of this paper is the novel conceptualization of image fusion as a dynamic, agent-based planning problem rather than a static pipeline. By employing a VLM as a controller, FuseAgent can explicitly reason about and adapt to the complex, coupled degradations found "in-the-wild". This is a significant paradigm shift from traditional methods, which are often brittle and fail to generalize to such varied and unpredictable inputs.
2. The two-stage training methodology (SFT + GRPO-F) is well-conceived and effectively executed. The SFT phase provides a strong initialization by mimicking expert trajectories , while the GRPO-F stage allows the agent to generalize and discover superior strategies on unlabeled data. The design of the unsupervised, multi-dimensional reward signal (IQR and RQR) is a notable contribution, enabling the agent to learn complex policies by balancing standalone perceptual quality with the critical "fusion compatibility" between source images. The ablation studies strongly support the value of this hybrid training approach and the combined reward structure.

**Weaknesses:**

1. The framework's performance appears to be heavily dependent on the comprehensiveness and quality of the expert models available in its library. While the authors state they used representative models, it's unclear how the agent would perform if it encounters a novel degradation for which no suitable tool is available. The paper could benefit from a discussion on the scalability of the tool suite and the agent's robustness to missing or inadequate tools.
2. The SFT stage relies on a dataset of "expert trajectories". The paper states that the "optimal action trajectory" for this dataset is determined using an "exhaustive search strategy". This approach may be computationally feasible for a limited number of tools and short action sequences, but its scalability is questionable. The paper does not fully address how this data generation pipeline would cope with a significant increase in the number of tools or the complexity of the required workflows.

**Questions:**

1. Regarding the tool suite: How does FuseAgent handle a scenario where it correctly identifies a degradation, but no appropriate "expert model" for that specific defect exists in its library?

2. The SFT data generation relies on an "exhaustive search" to define the optimal trajectory. Could the authors elaborate on the computational complexity of this search and its scalability as the number of expert models and potential trajectory length increase?

3. Could the authors comment on the inference cost of FuseAgent compared to other baselines? Further, how does the proposed method's efficiency suit real-world use cases?

---

> ### Author Response · Authors · 2025-11-21
>
> We are grateful for your insightful comments regarding the scalability and robustness of our framework.
>
> > Q1: Robustness to missing or inadequate tools.
>
>  FuseAgent demonstrates strong robustness through zero-shot tool transfer and fast adaptation capabilities.
>
> (1) **Zero-shot Tool Transfer:** FuseAgent can repurpose existing tools for new tasks via in-context learning. As shown in Table A, we tested an "Over-exposure" scenario where no explicit "Exposure Fixer" was defined in the training set. FuseAgent successfully identified the "Histogram Matching" tool (typically used for color consistency in MFF) as a valid substitute based on its textual description, significantly outperforming the baseline.
>
> Table A: Performance on New Degradation
>
> | Method                | **HyperIQA (↑)** | **CLIPIQA (↑)** | **BRISQUE (↓)** | **MUSIQ (↑)** |
> | --------------------- | ---------------- | --------------- | --------------- | ------------- |
> | ViT-Large（Baseline） | 0.2406           | 0.1394          | 28.4829         | 40.0946       |
> | CDDFuse               | 0.2983           | 0.1392          | 25.5609         | 48.5643       |
> | Text-IF               | 0.2576           | 0.1787          | 28.9608         | 42.0797       |
> | FILM                  | 0.2962           | 0.1646          | 27.5283         | 45.2466       |
> | FuseAgent             | 0.4836           | 0.2332          | 24.6227         | 55.0051       |
>
> (2) **Fast Adaptation via Fine-tuning:** Our framework supports efficient scalability. Adding a new tool does not require retraining the agent from scratch. It only involves three steps: (a) collecting a small set of unlabeled data, (b) adding the new tool description to the prompt, and (c) performing unsupervised fine-tuning with the GRPO-F algorithm. This allows the agent to rapidly encompass new capabilities with minimal computational cost.
>
> > Q2: Scalability of the exhaustive search.
>
> We clarify that our data generation does not rely on a brute-force $N!$ search.
>
> (1) **Pruned Space:** We rigorously prune the search space based on task applicability (e.g., ignoring de-raining tools for indoor MEF) and logical constraints (e.g., Registration must precede Fusion).
>
> (2) **Offline Cost:** This search is a one-time offline process used solely for generating the SFT dataset. With our current library, generating the optimal trajectory for one sample takes ~1.6s on parallel GPUs. This computational cost is negligible for dataset creation and has absolutely no impact on the model's online inference efficiency.
>
> >Q3: Inference cost.
>
> We provide a detailed comparison in Table B. By employing the 3B variant of our VLM, FuseAgent achieves an inference speed of 0.37s. This offers a superior trade-off between performance and efficiency compared to heavy all-in-one models like FILM (14.11s). As a pioneering effort, this study introduces the concept of an intelligent fusion agent to explore a new paradigm. We believe efficiency can be further enhanced through strategies such as (1) model quantization and (2) speculative sampling. We will continue to explore these acceleration techniques to refine this paradigm in future work.
>
> Table B: Inference Latency Comparison
>
> | Method                    | Text-IF | IF-MT-SSL | FILM   | FuseAgent (3B) | FuseAgent (7B) |
> | :------------------------ | :-----: | :-------: | ------ | -------------- | -------------- |
> | **Time (s)**              |  0.18s  |   0.30s   | 14.11s | 0.37s          | 1.55s          |
> | **HyperIQA ($\uparrow$)** | 0.4739  |  0.4521   | 0.5136 | 0.5683         | 0.6311         |

---

> ### Author Response · Authors · 2025-11-26
> **Response to Reviewer mtHR**
>
> Thanks for your questions.  Here are our clarifications regarding the tool semantics and the optimization of reasoning.
>
> > Q4: Language Priors and Learning Policy Interaction.
>
> Actually, the language serves as the bridge between visual perception and decision-making.  However, the agent relies on language differently depending on whether the tool is *known* or *novel*.
>
> (1) Known Tools (Training Stage).
>
> In this stage, experiential feedback dominates textual descriptions. Through SFT and RL, the agent learns by trial and error: when a tool successfully resolves a degradation type, the agent receives a high reward and thus forms a stable functional association. For example, it learns that “Tool A corrects Blur” even if the tool's textual definition is vague. As a result, the learned policy becomes robust to ambiguous or imprecise descriptions, because it is grounded in actual interaction experience rather than linguistic cues.
>
> (2) Unseen Tools (Zero-shot Transfer).
>
> In contrast, language priors become critical when the agent encounters tools it has never used. Lacking experiential data, the agent must depend on the textual definitions to infer each tool’s utility. Misleading descriptions can therefore cause planning failures. However, as long as the description captures the core functionality, the agent can generalize effectively. For example, in Q1, the agent infers from its SFT pre-training that “Histogram Matching” relates to *exposure correction*, enabling correct use despite having no direct training experience with this tool.
>
> > Q5:  Is CoT Reasoning Optimized with GRPO-F?
>
> The GRPO-F algorithm optimizes the model’s CoT reasoning by explicitly modeling reasoning as the causal driver of all subsequent actions. The workflow of tool-generated images follows the sequence: CoT reasoning → tool invocation decisions → final image → reward computation. The resulting reward serves as the optimization signal, allowing GRPO-F to update all tokens along the trajectory, including intermediate reasoning steps and tool-invocation tokens.
>
> For example, when the agent produces incorrect logic—such as hallucinating image degradation—this flawed reasoning leads directly to improper tool use. These unnecessary operations reduce image fidelity and thus diminish the intrinsic quality reward. Since reinforcement learning penalizes the full trajectory, it discourages not only the erroneous actions themselves but also the underlying reasoning patterns responsible for them.
>
> Additionally,  comparing the SFT and RL models, we observe two distinct shifts in the reasoning text: (1) **Narrow Hallucination:** The SFT model tends to copy human annotations and often over-diagnoses problems. The RL model becomes more pragmatic. It learns to identify degradations only when fixing them yields a tangible reward gain. (2) **Streamlined Logic:** The reasoning becomes clearer. It pays less attention to filler words and more attention to instrumental problems. The optimized direction is related to alignment (RQR) or quality (IQR). The agent learns to think in such a way as to bring it directly to a high-scoring outcome.

---

> > ### Comment · Reviewer_mtHR · 2025-11-27
> >
> > Thank you for your reply, I have no other questions.

---

> > > ### Author Response · Authors · 2025-11-27
> > >
> > > Dear Reviewer mtHR,
> > >
> > > We’re pleased that our rebuttal has addressed your concerns. We sincerely appreciate your insightful comments, which we believe will help us further enhance the clarity and depth of our work in the revision.
> > >
> > > Best regards, The Authors

---

### Official Review · Reviewer_Ad7Z · 2025-10-29

**Soundness:** 3
**Presentation:** 3
**Contribution:** 3
**Rating:** 4
**Confidence:** 5

**Summary:**

This paper introduces FuseAgent, a VLM-driven agent system designed to tackle the complex and highly variable problem of in-the-wild image fusion. The agent autonomously analyzes coupled degradations in the input images (e.g., misalignment, noise, dynamic artifacts) and dynamically plans a customized processing workflow composed of "expert models", and it is trained to cope with these problems by learning different expert models.

The groping method, combined with the fractal space and Gaussian-flat pattern, yields a clever double-layered training framework (SFT and GRPO-f), optimizes an unsupervised reward mechanism (IQR and RQR) and demonstrates a general automatic-tuning memory in excess of human experts. Moreover, the paper gives simulations to demonstrate its technical prowess.

**Strengths:**

**Novel Paradigm**: The primary contribution is introducing the VLM Agent paradigm to image fusion. This dynamic "Perceive-Plan-Execute" framework, compared to traditional static models, represents a paradigm-level advance for handling complex, heterogeneous degradation scenarios.

**Superior Decision-Making**: As shown in Table 1, FuseAgent's ability to plan the optimal processing path is impressive. Its performance surpasses not only zero-shot VLM planners and fixed, pre-defined pipelines but also the workflows manually designed by human experts, demonstrating powerful reasoning and planning capabilities.

**Performance**: On the challenging "in-the-wild" benchmark constructed by the authors, the authors claims that FuseAgent significantly outperforms existing task-specific and all-in-one SOTA methods on several key perceptual metrics (e.g., HyperIQA, CLIPIQA).

**Effective Training Framework**: The two-stage training strategy combining SFT and GRPO-F, along with the unsupervised IQR/RQR reward design, is very clever. It provides a viable and efficient blueprint for training agents to solve complex, multi-step vision tasks in the absence of expensive ground-truth labels.

**Weaknesses:**

**Questionable Premise of Unification**: The paper's core premise rests on building a "unified" fusion framework. However, different fusion tasks (e.g., Infrared-Visible vs. Multi-Exposure) are based on fundamentally different imaging principles and physics, $ESPECIALLY$ in facing an in-the-wild situation. The paper fails to sufficiently justify why forcing these physically dissimilar tasks into a single model is necessary, or what unique advantages this unification offers over specialized, domain-specific models. This makes the "unification" premise seem more like an artificial setup to inflate the paper's contribution (contribution), rather than a solution driven by a practical, real-world need.

**Fairness of Experimental Comparison**: The paper compares FuseAgent against existing SOTA methods (e.g., FILM, DRMF) on the authors' "in-the-wild" dataset. However, these SOTA methods were largely designed with the assumption of relatively clean or pre-aligned inputs; they were not built to handle such complex coupled degradations. Therefore, comparing them in a scenario they were not designed for is arguably unfair. A much more convincing baseline would be a "Preprocessing + SOTA model" pipeline (e.g., using the paper's own registration tool + the FILM fusion model). The current absence of this stronger, fairer baseline significantly weakens the paper's claim to SOTA performance.

**Lack of Efficiency and Latency Analysis**: Authors mentioned that the paper targets real-world applications like autonomous driving, which are often latency-sensitive, yet provides absolutely no analysis of the system's inference speed, end-to-end latency, or computational cost. However, the proposed multi-step, agent-driven framework (VLM planning + sequential execution of multiple expert models) inherently introduces significant computational overhead and latency.  This is a major practical limitation that is left unaddressed.

**Reliance on Off-the-Shelf Tools**: The paper's novelty lies entirely in the orchestration of expert tools, not in the tools themselves. The "expert models" (LoFTR, Restormer, SwinFusion, etc.) are all pre-existing, off-the-shelf methods, treated as black boxes. While this is a valid agent-based approach, it means the paper does not contribute any new, fundamental model architectures for the core tasks of preprocessing or fusion. The system's performance is therefore heavily capped by the capabilities of these external tools.

**Questions:**

As introduced in Weaknesses part, this is the following questions:

**Regarding Weakness #1 (Unification Premise)**: Could the authors elaborate on the practical or theoretical justification for creating a unified agent for physically dissimilar fusion tasks (like IVIF and MEF)? Beyond the goal of handling mixed degradations, what specific advantages does a single unified agent provide over using separate, specialized agents for each task?

**Regarding Weakness #2 (Experimental Fairness)**: This is the key question. Have the authors conducted, or can they provide, results from experiments using a stronger, more "apples-to-apples" baseline, such as a "Preprocessing + SOTA Model" pipeline? For example, running the alignment tools from the agent's tool-suite on the inputs first, and then feeding the aligned images to a SOTA method like FILM or DRMF. How would FuseAgent's performance compare against such a baseline?

**Regarding Weakness #3 (Efficiency)**: Could the authors provide any benchmark or estimation of the system's end-to-end latency (VLM planning + all model executions)? How does this computational cost compare to a single end-to-end SOTA fusion model, and how do the authors envision this being deployed in the latency-sensitive applications mentioned?

**Regarding Weakness #4 (Tool Reliance)**: Given that the system's performance is bound by the quality of the expert tools, did the authors experiment with replacing or fine-tuning these tools? For instance, how much of the performance gain comes from the orchestration itself, versus simply being the first to apply a superior (but pre-existing) registration model like LoFTR within a fusion pipeline?

---

> ### Author Response · Authors · 2025-11-21
> **Rebuttal Part 1**
>
> We appreciate your positive feedback on our clear motivation and the comprehensiveness of our framework.
>
> > Q1: Justification for the "Unification" premise.
>
> We clarify that the rationale for unification is grounded in the inherent complexity of in-the-wild environments, where the theoretical boundaries between distinct tasks, such as IVIF versus MEF, become increasingly ambiguous.
>
> (1) **Convergence of Real-World Challenges:** Although the underlying imaging principles differ, real-world scenarios often present shared degradation patterns across tasks, such as geometric misalignment, motion artifacts, and extreme lighting. For instance, "ghosting" caused by object motion in IVIF scenes presents similar challenges to those found in MEF. A unified agent can abstract these common problems into generalized skills (e.g., *Semantic Alignment*, *Dynamic Artifact Removal*), allowing it to apply effective solutions universally across different domains.
>
> (2) **Cross-Task Capability Transfer:** Unification enables the flexible application of domain-specific tools across different tasks. To validate this, we compared our method against a strong baseline consisting of a ViT-Large classifier with fixed rules on a specific "Combined Degradation" dataset of approximately 150 samples. This dataset features cross-task challenges, such as IVIF images compounded with heavy fog and night scene. As shown in Table A, the specialized baseline fails to adapt to these complex cases. In contrast, our Unified Agent successfully transfers restoration capabilities to the fusion task, demonstrating superior robustness.
>
> Table A: Robustness on Cross-Task Degradations
>
> | Evaluation Scenario                                          | ViT-Large (Baseline) | FuseAgent (Ours) |
> | :----------------------------------------------------------- | :------------------: | :--------------: |
> | **Combination Degradation***(Metric: HyperIQA $\uparrow$)* |        0.4829        |    **0.5730**    |
>
> > Q2: Fairness of experimental comparison.
>
> (1) **Clarification on Original Setup:** We respectfully highlight that **Section 4.2** of our manuscript explicitly addresses this comparison via the "Pre-defined Order & Model" baseline. As detailed in the paper, this strategy employs a fixed sequence of expert tools (e.g., Restoration → Registration) prior to fusion, effectively representing the "Preprocessing + SOTA" pipeline you suggested.
>
> (2) **Evaluation on Augmented Baseline:** To further validate this, we implemented the exact pipeline you requested using the FILM model, as detailed in Table B. We integrated the same registration and restoration tools used by FuseAgent into the FILM baseline, governed by a standard fixed logic (Row 2). As shown in Table B, while this "Augmented Baseline" (Row 2) improves upon the vanilla FILM (Row 1), it remains significantly inferior to FuseAgent (Row 4). This empirical evidence demonstrates that tool availability alone is insufficient; the adaptive, context-aware orchestration provided by the agent is the decisive factor for achieving SOTA performance.
>
> Table B: Decision Making Strategies
>
> | Strategy                  | Planner       | Preprocessing | Fusion Model | HyperIQA ($\uparrow$) |
> | :------------------------ | :------------ | :------------ | :----------- | :-------------------: |
> | **(1) FILM (Baseline)**   | N/A           | None          | FILM         |        0.5136         |
> | **(2) Fixed-Rule + FILM** | Fixed Rule    | Fixed Tools   | FILM         |        0.5539         |
> | **(3) FuseAgent + FILM**  | **FuseAgent** | **Adaptive**  | FILM         |        0.5808         |
> | **(4) FuseAgent (Ours)**  | **FuseAgent** | **Adaptive**  | **Adaptive** |      **0.6311**       |

---

> > ### Author Response · Authors · 2025-11-21
> > **Rebuttal Part 2**
> >
> > > Q3: Efficiency and latency analysis.
> >
> > Table C summarizes the inference efficiency. We evaluated latency using images with a resolution of 512×512 on a single A100 GPU. By employing the 3B variant of our VLM, FuseAgent achieves a total inference time of 0.37s, offering a superior trade-off between performance and efficiency. Specifically, it is comparable in speed to lightweight methods like IF-MT-SSL (0.30s) while delivering significantly higher robustness (HyperIQA 0.5683 vs 0.4521), and is orders of magnitude faster than heavy baselines like FILM (14.11s). To address latency-sensitive scenarios like autonomous driving, we propose a **"Plan-Execute" decoupling strategy**: the heavy VLM planning runs only upon scene changes (e.g., entering a tunnel), while the lightweight tool execution runs per frame, ensuring real-time performance. As a pioneering effort, this work introduces a novel agentic paradigm capable of handling complex degradations via reasoning. We believe efficiency can be further enhanced through strategies such as model quantization and speculative sampling, which we plan to explore in future work.
> >
> > Table C: Inference Latency Comparison
> >
> > | Method                    | Text-IF | IF-MT-SSL | FILM   | FuseAgent (3B) | FuseAgent (7B) |
> > | :------------------------ | :-----: | :-------: | ------ | -------------- | -------------- |
> > | **Time (s)**              |  0.18s  |   0.30s   | 14.11s | 0.37s          | 1.55s          |
> > | **HyperIQA ($\uparrow$)** | 0.4739  |  0.4521   | 0.5136 | 0.5683         | 0.6311         |
> >
> > > Q4: Reliance on off-the-shelf tools.
> >
> > Our core contribution is the intelligent agent framework itself, which introduces a fundamental paradigm shift from "tool optimization" to "autonomous perception and reasoning."
> >
> > (1) **Divergent Research Paradigms:** We emphasize that Agentic Intelligence and Tool Development serve fundamentally different roles. While tools focus on functional utility (the "Hands"), our work focuses on the cognitive layer (the "Brain"). FuseAgent treats image fusion as a **System-2 reasoning problem**, autonomously decomposing complex, coupled degradations into logical sub-tasks. This mirrors recent breakthroughs in other domains: just as Search-o1 [1] achieves reliability through autonomous reasoning chains rather than better search engines, and UI-TARS [2] achieves SOTA in GUI interaction via "System-2 Reasoning" rather than new drivers, FuseAgent leverages cognitive planning to solve complex vision tasks where static tools fail.
> >
> > (2) **Quantifiable Gain:** To explicitly isolate the contribution of FuseAgent's reasoning and planning capability, we present the attribution analysis in Table B. While utilizing the exact same toolset under a fixed-rule policy yields a score of 0.5539 (Row 2), activating FuseAgent's dynamic planning boosts performance to 0.5808 (Row 3). This gain is strictly attributable to the agent's reasoning and orchestration, proving that the *decision-making capability* is the decisive factor.
> >
> > ------
> >
> > References:
> >
> > [1] Search-o1: Agentic Search-Enhanced Large Reasoning Models. EMNLP 2025.
> >
> > [2] UI-TARS: Pioneering Automated GUI Interaction with Native Agents. arXiv 2025.

---

> ### Author Response · Authors · 2025-11-27
>
> Dear Reviewer Ad7Z,
>
> I hope this message finds you well. As the discussion period is nearing its end, I wanted to ensure we have addressed all your concerns satisfactorily. If there are any additional points or feedback you'd like us to consider, please let us know. Your insights are invaluable to us, and we're eager to address any remaining issues to improve our work.
>
> Thank you for your time and effort in reviewing our paper.

---

### Official Review · Reviewer_bKw8 · 2025-11-01

**Soundness:** 2
**Presentation:** 3
**Contribution:** 2
**Rating:** 4
**Confidence:** 5

**Summary:**

The authors propose FuseAgent, a novel agentic system for multi-source image fusion designed to handle complex, real-world degradations. This system employs a Vision-Language Model as a central controller to identify image distortions and dynamically assemble a processing pipeline from a set of expert models. The agent is trained via a two-stage process, beginning with Supervised Fine-Tuning on expert trajectories, followed by a reinforcement learning phase using a custom algorithm, Group Relative Policy Optimization for Fusion. To facilitate this second stage, the authors introduce two unsupervised reward metrics—the Intrinsic Quality Reward and the Relational Quality Reward to guide the agent's policy development. The paper claims that this approach achieves significant performance improvements over existing methods on in-the-wild datasets.

**Strengths:**

The paper's main strength lies in its conceptualization of the image fusion problem from an agentic perspective, which is an new direction for low-level vision tasks.

**Weaknesses:**

1. The authors claim that the VLM is capable of "perceiving" degradations. It is unclear how this "perception" fundamentally differs from the outcome of a standard classification model trained to identify degradation types. Could the authors elaborate on the unique advantages or novel capabilities that the VLM brings to this specific task compared to a more straightforward classification approach?
2. The fusion pipeline detailed in Section 3.1 appears extensive, which suggests it may incur substantial computational overhead and GPU memory consumption. To properly evaluate its practical efficiency, the authors should provide a detailed comparison of FuseAgent's training time, inference time, and memory footprint against the baseline methods.
3.  For the metrics:
    a) The authors used a series of IQA metrics during the training of the agent. However, these same metrics were also used for comparison in the evaluation stage. Is this comparison fair to the other methods? To my knowledge, most of the compared methods do not rely on IQA-based metrics for training, validation or testing.
    b) Moreover, the BRISQUE metric used during training is not even included in the comparisons in Table 3. SSIM and SF, which also appeared during the agent’s training, are again used as evaluation metrics. Therefore, all the metrics compared in this paper are those already used in the training process.
    c) In addition, the metrics used across different tasks in Tables 2 and 3 are inconsistent, and SSIM does not appear in Table 1. These issues require further clarification from the authors.
4. Following the previous point, the methodology seems to tightly couple the fusion process with a curated set of evaluation metrics. What were the specific criteria for selecting these reward metrics? A clear justification is needed for why certain metrics (e.g., SF) were included while other common fusion metrics (e.g., EN) were omitted. Furthermore, on line 332, the paper claims that metrics like Qabf, Qcb, and VIF become unreliable for degraded images. Could the authors provide a rationale for why SSIM, which also has known limitations with severe degradations, is considered exempt from this issue?
5. In the comparative experiments on degraded images, it is unclear whether the baseline fusion methods were originally designed to handle such complex source image degradations. If not, the comparison lacks fairness. If the baselines were augmented with external degradation or misalignment-handling modules for the sake of comparison, this introduces a confounding variable. In that case, how can the authors definitively attribute the reported performance gap to the fusion algorithm itself, rather than the potential inadequacy of the added pre-processing modules? The experimental setup must be clarified to ensure a conclusive and fair comparison.

**Questions:**

Please see Weaknesses section

---

> ### Author Response · Authors · 2025-11-21
> **Rebuttal Part 1**
>
> We sincerely appreciate your constructive feedback and recognition of the novelty of applying VLM agents to image fusion.
>
> > Q1: Perception (VLM) vs. Standard Classification.
>
> The "perception" of a VLM fundamentally differs from a standard classifier's closed-set mapping. While classifiers assign inputs to fixed labels, FuseAgent leverages VLM's reasoning to interpret complex degradations, offering three key advantages:
>
> 1.  **Open-Set Generalization:** VLMs use semantic understanding for zero-shot recognition of novel defects (e.g., "over-exposure"), whereas classifiers fail on unseen classes.
> 2.  **Compositional Reasoning:** VLMs decompose entangled degradations (e.g., "fog + misalignment") into logical sub-problems, overcoming the multi-label ambiguity faced by classifiers.
> 3.  **Tool-Aware Planning:** VLMs dynamically map defects to tool functions based on textual descriptions, enabling the use of tools not explicitly taught during training.
>
> Table A compares FuseAgent against a strong baseline (ViT-Large classifier + fixed rules) across four scenarios. While the baseline is competitive in-domain, it fails on **Unseen Scenes** (Scenario 2, HyperIQA: 0.45 vs. 0.52) and **New Degradations** (Scenario 4, 0.24 vs. 0.48). Notably, in Scenario 4, the ViT misclassified the unknown artifact, whereas FuseAgent successfully reasoned that the "Histogram Matching" tool could be repurposed to fix exposure. This confirms that VLMs provide unique **reasoning and transfer capabilities** beyond simple classification.
>
> Table A: Quantitative comparison with a strong baseline across four generalization scenarios.
>
> | Evaluation Scenario           | **Method**    | **HyperIQA (↑)** | **CLIPIQA (↑)** | **MUSIQ (↑)** | **BRISQUE (↓)** |
> | ----------------------------- | ------------- | ---------------- | --------------- | ------------- | --------------- |
> | **1. In-Domain Benchmark**    | ViT-Large     | 0.5809           | 0.3553          | 55.6211       | 25.1137         |
> | *(Baseline on our Test Set)*  | **FuseAgent** | **0.6309**       | **0.4014**      | **63.9540**   | **18.1184**     |
> | **2. Unseen Scene**           | ViT-Large     | 0.4548           | 0.2941          | 50.3079       | 41.8958         |
> | *(Test for New Environment)*  | **FuseAgent** | **0.5233**       | **0.4146**      | **59.5347**   | **38.2772**     |
> | **3. Compound Degradation**   | ViT-Large     | 0.4829           | 0.2826          | 51.5781       | 17.3043         |
> | *(Test for Tool Composition)* | **FuseAgent** | **0.5730**       | **0.3804**      | **59.2648**   | **15.3786**     |
> | **4. New Degradation & Tool** | ViT-Large     | 0.2406           | 0.1394          | 40.0946       | 28.4829         |
> | *(Test for Tool Discovery)*   | **FuseAgent** | **0.4836**       | **0.2332**      | **55.0051**   | **24.6227**     |
>
> **Note:**
>
> *   **In-Domain:** Standard held-out test set.
> *   **Unseen Scene:** Generalization to novel environments (EMS dataset [1]).
> *   **Combined Degradation:** Unseen compound degradations (e.g., "Night + Fog + Misalignment").
> *   **New Degradation & Tool:** Zero-shot tool discovery for over-exposure.

---

> > ### Author Response · Authors · 2025-11-21
> > **Rebuttal Part 2**
> >
> > > Q2: Computational overhead and efficiency.
> >
> > Table B summarizes the training costs and inference efficiency. We evaluated inference latency using 200 images with a resolution of $512\times512$ on a single A100 GPU. Results show that FuseAgent-7B achieves a total inference time of 1.55s, significantly outperforming the leading diffusion-based solution, UltraFusion (14.29s), and approaching the speed of TC-MoA (0.67s). Furthermore, by switching the backbone to the Qwen2.5-VL-3B variant with shorter reasoning, the total inference time is reduced to 0.37s, making it comparable to lightweight models like IF-MT-SSL (0.30s) and Text-IF (0.19s). As a pioneering effort, this work introduces the concept of an intelligent fusion agent to establish a new research paradigm. We believe efficiency can be further enhanced through strategies such as model quantization and speculative sampling, which we plan to explore in future work.
> >
> > Table B: Comparison of training cost and inference efficiency across different methods.
> >
> > | Method       | Training Time | Training VRAM Usage | Inference Time                               | Inference VRAM Usage |
> > | ------------ | ------------- | ------------------- | -------------------------------------------- | -------------------- |
> > | Text-IF      | 60 GPU hours  | ～32GB              | 0.19s                                        | ~5GB                 |
> > | FILM         | 161 GPU hours | ～61GB              | 14.11s (includes caption and embedding time) | ~7GB                 |
> > | TC-MoA       | 36 GPU hours  | ～70GB              | 0.67s                                        | ~3GB                 |
> > | UltraFusion  | 428 GPU hours | ～48GB              | 14.29s                                       | ~11GB                |
> > | IF-MT-SSL    | 271 GPU hours | ～50GB              | 0.303s                                       | ~2GB                 |
> > | FuseAgent-3B | 30 GPU hours  | ～60GB              | 0.37s                                        | ~8GB                 |
> > | FuseAgent    | 384 GPU hours | ~140GB              | 1.55s                                        | ~13GB                |
> >
> >
> > > Q3: Fairness of metrics.
> >
> > We acknowledge the concern regarding potential metric overlap. However, using non-reference IQA models as reward signals is a necessity for **label-free RL**. To verify that our evaluation is fair and that FuseAgent is not simply overfitting to these specific metrics, we conduct two additional experiments:
> >
> > (1) **Generalization to Unseen Evaluators (Table C):** We evaluated FuseAgent on metrics **never seen during training** (e.g., Q-Align, TOPIQ, LIQE). FuseAgent consistently achieves SOTA results (e.g., Q-Align: 2.68 vs. 2.47 for CDDFuse), indicating that RL training improves genuine perceptual quality rather than exploiting metric-specific artifacts.
> >
> > (2) **Metric Validity Verification (Table D):** To verify that the chosen metrics do not unfairly penalize baseline methods, we evaluate all models on a "Clean Benchmark". Baselines perform on par with FuseAgent under these non-degraded conditions (e.g., CLIPIQA: 0.46 vs. 0.45 for FILM), showing that the metrics are unbiased and valid quality indicators. The large performance gaps in degraded cases (Table 2,3) therefore reflects the baselines' genuine inability to handle degradations, not metric bias.
> >
> > Table C: Generalization to New Metrics
> >
> > | Method    | BRISQUE(↓) | Q-Align(↑) | PaQ-2-PiQ(↑) | LIQE(↑) | PIQE(↓) | TReS(↑) | TOPIQ(↑) |
> > | --------- | ---------- | ---------- | ------------ | ------- | ------- | ------- | -------- |
> > | Text-IF   | 17.374     | 2.4137     | 71.6443      | 1.8667  | 34.2072 | 68.9145 | 0.4521   |
> > | FILM      | 17.5008    | 2.3323     | 70.5395      | 2.0527  | 31.5823 | 74.5597 | 0.5082   |
> > | IF-MT-SSL | 18.1314    | 2.2374     | 66.9171      | 1.456   | 25.2914 | 61.6159 | 0.4076   |
> > | CDDFuse   | 13.6354    | 2.478      | 69.8063      | 1.842   | 26.7937 | 68.2782 | 0.4604   |
> > | FuseAgent | 16.6587    | 2.6869     | 73.1415      | 2.9316  | 21.2908 | 80.48   | 0.5755   |
> >
> >
> > Table D: Baseline Performance on Clean Data
> >
> > | Method    | EN(↑)  | SF(↑)  | MS-SSIM(↑) | HyperIQA(↑) | BRISQUE(↓) | MUSIQ(↑) | CLIPIQA(↑) | Q-Align(↑) |
> > | --------- | ------ | ------ | ---------- | ----------- | ---------- | -------- | ---------- | ---------- |
> > | Text-IF   | 7.3443 | 8.5759 | 0.5408     | 0.5664      | 14.0322    | 65.1928  | 0.3985     | 3.0394     |
> > | FILM      | 7.3291 | 8.3341 | 0.6484     | 0.6107      | 16.9303    | 64.4579  | 0.4535     | 3.0199     |
> > | IF-MT-SSL | 6.6691 | 7.1065 | 0.6755     | 0.5463      | 12.8991    | 58.1103  | 0.4177     | 2.8987     |
> > | CDDFuse   | 7.2303 | 7.9923 | 0.6693     | 0.5862      | 16.7343    | 65.235   | 0.3730     | 3.0351     |
> > | FuseAgent | 7.1924 | 7.7800 | 0.6685     | 0.6641      | 17.6644    | 67.6377  | 0.4698     | 3.0908     |

---

> > > ### Author Response · Authors · 2025-11-21
> > > **Rebuttal Part 3**
> > >
> > > > Q4: Metric selection criteria and SSIM.
> > >
> > > The selection of reward vs. evaluation metrics follows distinct criteria:
> > >
> > > (1) **Reward Design**: We selected metrics that provide dense, robust feedback for label-free RL optimization. Metrics like SF (Spatial Frequency) offer structural gradients, while HyperIQA aligns with perceptual quality. We omitted EN (Entropy) as it is often unstable and can be easily maximized by noise rather than valid information.
> > >
> > > (2) **Rationale for SSIM:**
> > >
> > > - Evaluation: In **Table 1**, we did not use standard SSIM. For IVIF/MFF tasks, we used **MS-SSIM** (Multi-Scale SSIM) because severe degradations often corrupt fine-scale details while preserving coarse structures; MS-SSIM averages across scales, offering greater robustness than single-scale SSIM. For the MEF task, we used **MEF-SSIM$_d$** [2], which is explicitly designed to handle dynamic motion artifacts (ghosting) that standard SSIM treats as errors. This difference in metric definitions prevents averaging them into a single "SSIM" column in Table 1.
> > > - Training: We employ SSIM within the *Relational Quality Reward* solely to assess inter-image compatibility, *not* final fusion quality. Unlike the standard evaluation metric which compares the fusion result to inputs (i.e., $avg(SSIM(I_A,I_F),SSIM(I_B,I_F))$), our reward measures the structural correlation between the *processed* source images (e.g., $A_i(I_A)$ vs. $A_i(I_B)$) across the N rollout candidates {${A_1, A_2, ..., A_N}$}. This explicitly encourages the agent to select trajectories that maximize the mutual structural consistency (alignment and exposure matching) between the two sources before the final fusion step.
> > >
> > > (3) Missing Metrics: We have added the missing BRISQUE scores to Table C (FuseAgent: 16.66 vs. Text-IF: 17.37, lower is better) to ensure completeness.
> > >
> > > > Q5: Fairness of baselines
> > >
> > > (1) **Robustness of Selected Baselines:** We clarify that several of the selected state-of-the-art baselines are inherently designed to handle complex degradations. for instance, Text-IF incorporates semantic guidance for degradation awareness, UMF-CMGR handles misalignment, and UltraFusion leverages generative priors for high dynamic range scenarios.
> > >
> > > (2) **Attributing Performance to the FusionAgent:** To isolate the contribution of the agent itself, we conducted detailed decoupling experiments, as reported in Table 1 of our manuscript and Table E. Specifically, we constructed an “augmented baseline” by equipping FILM with the same registration and restoration tools used by FuseAgent, but operating under a fixed-rule pipeline. Although this setup improves FILM’s performance (HyperIQA 0.51 $\to$ 0.55), it remains substantially below the FuseAgent-driven workflow (HyperIQA 0.63). These results demonstrate that the performance gains stem from FuseAgent’s adaptive decision-making rather than from the mere availability of additional tools.
> > >
> > > Table E: Decoupling the contribution of the Planner (Agent) vs the Tools.
> > >
> > > | **Strategy**               | **Planner**   | **Preprocessing Model** | **Fusion Model** | **HyperIQA (↑)** | **MUSIQ (↑)** | **CLIPIQA (↑)** | **BRISQUE (↓)** |
> > > | -------------------------- | ------------- | ----------------------- | ---------------- | ---------------- | ------------- | --------------- | --------------- |
> > > | **(1) Baseline**           | N/A           | None                    | FILM             | 0.5136           | 59.5164       | 0.2956          | 17.9008         |
> > > | **(2) Augmented Baseline** | Fixed Rule    | Fixed Tools             | FILM             | 0.5539           | 59.9219       | 0.2941          | 20.7248         |
> > > | **(3) Agent-Guided FILM**  | **FuseAgent** | **Adaptive**            | FILM             | 0.5808           | 61.3893       | 0.3335          | 17.1841         |
> > > | **(4) FuseAgent (Ours)**   | **FuseAgent** | **Adaptive**            | **Adaptive**     | **0.6311**       | **64.3891**   | **0.4122**      | **16.6587**     |
> > >
> > > ------
> > >
> > > References:
> > >
> > > [1] Text-if: Leveraging semantic text guidance for degradation-aware and interactive image fusion. CVPR 2024.
> > >
> > > [2] Perceptual evaluation for multi-exposure image fusion of dynamic scenes. TIP 2020.

---

> ### Author Response · Authors · 2025-11-27
>
> Dear Reviewer bKw8,
>
> I hope this message finds you well. As the discussion period is nearing its end, I wanted to ensure we have addressed all your concerns satisfactorily. If there are any additional points or feedback you'd like us to consider, please let us know. Your insights are invaluable to us, and we're eager to address any remaining issues to improve our work.
>
> Thank you for your time and effort in reviewing our paper.

---

### Meta-Review · Area_Chair_x84z · 2026-01-06

**Summary:**

This paper proposes FuseAgent, a VLM-driven agent that “perceives–plans–executes” by selecting and composing a library of expert models to address coupled degradations in in-the-wild multi-source image fusion, trained via SFT followed by a GRPO-F reinforcement learning stage with multi-dimensional rewards. The reviewers agree the agentic formulation is interesting and potentially impactful for low-level vision. However, the submission’s claims are not yet supported with sufficiently convincing evidence and framing for ICLR. The main blockers are: (i) an insufficiently justified “unification” premise across physically dissimilar fusion tasks and unclear practical need, (ii) concerns about evaluation validity and fairness (heavy reliance on no-reference IQA metrics and an in-the-wild benchmark constructed by the authors, plus incomplete apples-to-apples baselines across tasks), and (iii) remaining questions about real-world deployability and sensitivity to the tool library (robustness when tools are missing/insufficient and scalability of the expert-trajectory generation procedure). Therefore, I decide to reject this paper.

**Reviewer Concerns:**

The main blockers are: (i) an insufficiently justified “unification” premise across physically dissimilar fusion tasks and unclear practical need, (ii) concerns about evaluation validity and fairness (heavy reliance on no-reference IQA metrics and an in-the-wild benchmark constructed by the authors, plus incomplete apples-to-apples baselines across tasks), and (iii) remaining questions about real-world deployability and sensitivity to the tool library (robustness when tools are missing/insufficient and scalability of the expert-trajectory generation procedure)

**Reviewer Scores:**

Had the reviewer been able to fully participate in the discussion, I believe their score would likely have remained largely unchanged.  I appreciate the feedback provided and will carefully address these points in a revised version of the manuscript.

---

### Decision · Program_Chairs · 2026-01-26

Reject